# CellAgent: LLM-Driven Multi-Agent Framework for Natural Language-Based Single-Cell Analysis

**Yihang Xiao**[1*], **Jinyi Liu**[2*], **Yan Zheng**[2*], **Shaoqing Jiao**[1,3*], **Jianye Hao**[2†], **Xiaohan Xie**[1],
**Mingzhi Li**[2], **Ruitao Wang**[2], **Fei Ni**[2], **Yuxiao Li**[2], **Zhen Wang**[4], **Xuequn Shang**[3], **Zhijie Bao**[1],
**Changxiao Yang**[1], **Jiajie Peng**[1,3†]

[1]AI for Science Interdisciplinary Research Center, School of Computer Science,
Northwestern Polytechnical University
[2]College of Intelligence and Computing, Tianjin University
[3]Key Laboratory of Big Data Storage and Management, Ministry of Industry and Information
Technology, Northwestern Polytechnical University
[4]School of Cybersecurity, Northwestern Polytechnical University

`jianye.hao@tju.edu.cn, jiajiepeng@nwpu.edu.cn`

## Abstract

Single-cell RNA sequencing (scRNA-seq) and spatial transcriptomics (ST) data analysis are pivotal for advancing biological research, enabling precise characterization of cellular heterogeneity. However, existing analysis approaches require extensive manual programming and complex tool integration, posing significant challenges for researchers. To address this, we introduce CellAgent, an autonomous, LLM-driven approach that performs end-to-end scRNA-seq and spatial transcriptomics data analysis through natural language interactions. CellAgent employs a multi-agent hierarchical decision-making framework, simulating a "deep-thinking" workflow to ensure that analytical steps are logically coherent and aligned with the overarching research goal. To further enhance its capabilities, we develop sc-Omni, a high-performance, expert-curated toolkit that consolidates essential tools for scRNA-seq and spatial transcriptomics analysis. Additionally, we introduce a self-reflective optimization mechanism, enabling automated, iterative refinement of results through specialized evaluation methods, effectively replacing traditional manual assessments. Benchmarking against human experts demonstrates that CellAgent achieves significant improvement in efficiency across multiple downstream applications while maintaining excellent performance comparable to existing approaches and preserving natural language interactions. By translating high-level scientific questions into optimized computational workflows, CellAgent represents a step toward a new, more accessible paradigm in bioinformatics, allowing researchers to perform complex data analyses autonomously. In lowering technical barriers, CellAgent serves to advance the democratization of the scientific discovery process in genomics.

## 1 INTRODUCTION

In recent years, the rapid development of technologies such as single-cell RNA sequencing (scRNA-seq) and spatial transcriptomics (ST) has revolutionized molecular biology, enabling scientists to investigate biological systems with unprecedented precision and depth (Luecken et al., 2025; Heumos et al., 2023b). These biological technologies have produced many large-scale datasets that require advanced computational tools to extract meaningful biological information efficiently.

---

[*]denotes equal contribution. [†]Corresponding authors.

Although current tools have significantly improved the efficiency and accuracy of scRNA-seq/ST data analysis (Sethi et al., 2024; Szałata et al., 2024), the manual and flexible application of various tools to analyze complex and massive single-cell sequencing data carries high labor costs. For example, researchers need to meticulously choose the appropriate tools, while fine-tuning appropriate hyperparameters that are suited to the unique characteristics of the input data (Lukas et al., 2023). This requirement for dual expertise in both computational programming and fundamental biology increases the complexity and cost of performing single-cell data analysis tasks (Zhou et al., 2024), creating a significant barrier that inhibits the discovery of biological mechanisms. Therefore, there is an urgent need for a more automated, natural language interaction-based, and functionally integrated analysis tool, which may enable users to direct complex analyses through natural language, thereby reducing technical barriers, improving data processing efficiency, and further promoting knowledge discovery with single-cell RNA-seq and spatial transcriptomics technologies.

To address these challenges, we introduce **CellAgent**, a multi-agent hierarchical framework (Figure 1) that simulates a "deep-thinking" workflow to ensure coherent, goal-oriented task execution. This framework is orchestrated by three specialized agents: a high-level Planner that decomposes complex user requests into manageable subtasks, an Executor that carries out the analysis by generating and running code, and an Evaluator that assesses the quality of the results. The efficacy of this architecture is then built upon several core innovations designed to work in synergy. First, to ground the analytical capabilities of LLMs in the domain-specific complexities of genomics, we developed sc-Omni, a comprehensive toolkit integrating expert knowledge and a wide array of analysis tools (Table 2). Second, the Evaluator drives a self-reflective optimization mechanism, which leverages automated evaluation methods for various analysis tasks to iteratively refine outcomes, replacing subjective manual assessments. Finally, to ensure seamless cooperation, we design a global and local integrative memory control mechanism for systematic storage and efficient retrieval of historical information, optimizing overall task performance. Compared to traditional tools, CellAgent supports natural language interaction and enables fully automated, unattended task execution.

Extensive benchmarking demonstrates that CellAgent's performance in task completion, quality, and efficiency, is comparable to that of human experts, and in certain aspects, it even surpasses human experts. Specifically, CellAgent achieves an average execution success rate of more than 96% across over 60 datasets. Furthermore, CellAgent demonstrates competitive performance across various downstream tasks, such as cell-type annotation, batch correction, trajectory inference, spatial domain identification, and spatial transcriptomics imputation. In summary, CellAgent achieves state-of-the-art results on several key bioinformatics tasks, all driven through the dialogue-based manner as in Figure 2. To promote collaboration and facilitate efficient single-cell RNA sequencing and spatial transcriptomics data analysis, we provide an interactive online platform for CellAgent, enabling seamless integration with the broader research community.

## 2 RELATED WORK

Recently, the impressive capabilities of LLMs have spurred the development of LLM-driven autonomous AI agents, which have successfully automated tasks across various scientific domains. For instance, ChatMOF (Kang & Kim, 2024) is an agent designed for materials science that leverages LLMs to extract key information from natural language inputs to perform tasks such as data retrieval, property prediction, and the generation of metal-organic framework (MOF) structures. Similarly, ChemCrow (M. Bran et al., 2024) acts as a chemistry agent, augmenting an LLM by integrating a suite of expert-designed tools to autonomously plan and execute complex organic synthesis and drug discovery workflows.

Although computational frameworks such as Scanpy (Wolf et al., 2018), Squidpy (Palla et al., 2022), Seurat (Hao et al., 2021), and scVI (Lopez et al., 2018), together with single-cell foundation models like scGPT (Cui et al., 2024) and cellPLM (Wen et al., 2023), have substantially advanced scRNA-seq and spatial transcriptomics analysis, they still require domain experts to manually construct complex code pipelines. In parallel, general-purpose LLMs such as GPT-4 and agentic frameworks like AutoGen (Wu et al., 2023) lack the domain-specific knowledge needed for robust biological analysis.

Recent efforts have explored complementary directions in developing LLM-driven agents for biomedical research. Biomni (Huang et al., 2025), a general-purpose biomedical agent that inte-

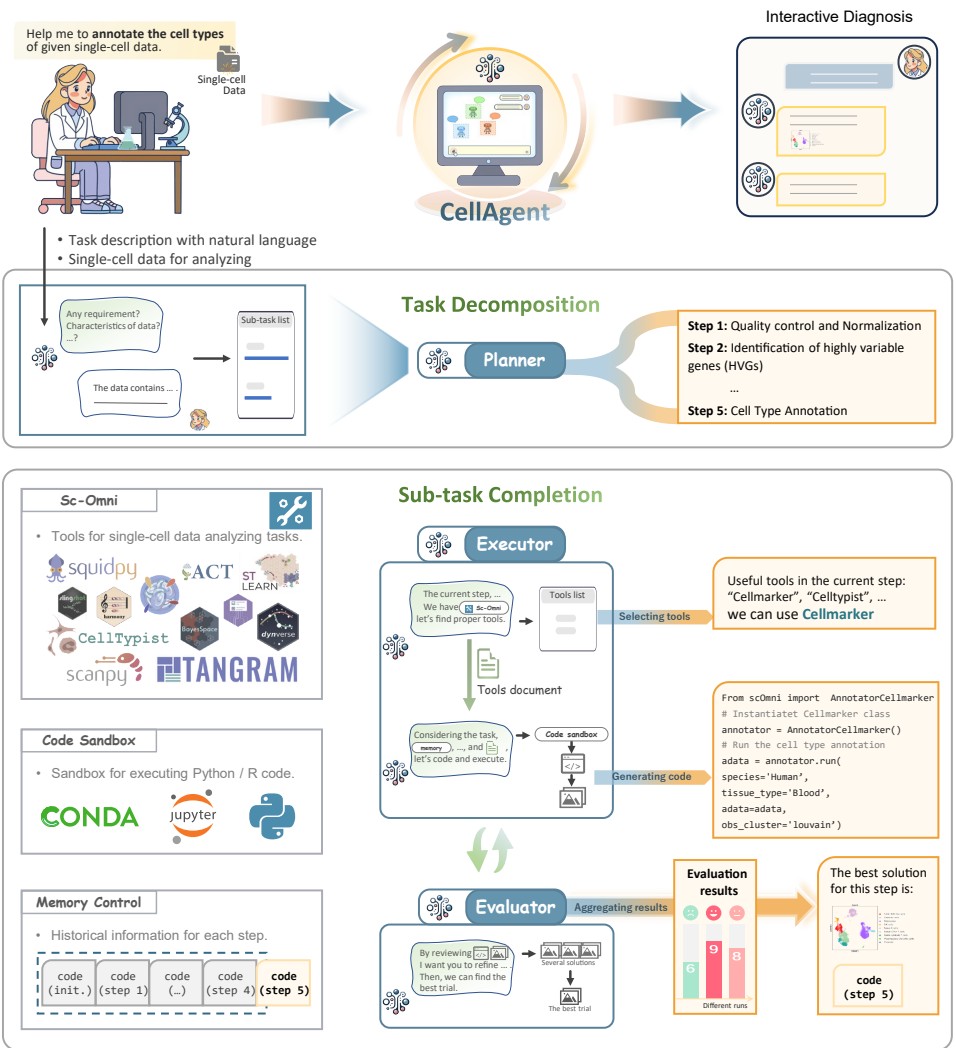

Figure 1: **Schematic of the CellAgent Framework.** Users interact with CellAgent via natural language to obtain high-quality, automated analysis results tailored to their specific needs. Then the framework operates hierarchically, with a high-level Planner that performs fine-grained task decomposition based on input data characteristics and user queries. In the lower-level execution phase, subtasks are completed sequentially. An Executor selects optimal tools from the sc-Omni toolkit to generate and execute code. An Evaluator then rigorously assesses the outcomes, proposing refinements if needed. This self-reflective optimization loop iterates to enhance precision, and the final results from all subtasks are synthesized to meet the user's requirements.

grates LLM reasoning with retrieval-augmented planning and code execution, facilitates the efficient formulation and empirical validation of scientific hypotheses. In parallel, domain-specialized agents such as AutoBA (Zhou et al., 2024) and BioMANIA (Dong et al., 2023) aim to lower programming barriers and automate single-cell data analysis. However, these approaches primarily focus on improving task execution success rates rather than enhancing analysis quality. A critical limitation remains the absence of mechanisms to automatically evaluate the biological relevance of results, which constrains their ability to autonomously select suitable algorithms and optimize hyperparameters across heterogeneous datasets, a challenge that has been consistently emphasized in prior studies (Heumos et al., 2023a; Rao et al., 2021). In contrast, CellAgent addresses this gap by explicitly integrating biological relevance assessment into its automated workflow through a collaborative architecture, where a dedicated Evaluator Agent and a self-reflective optimization mechanism jointly ensure high-quality, biologically meaningful outcomes.

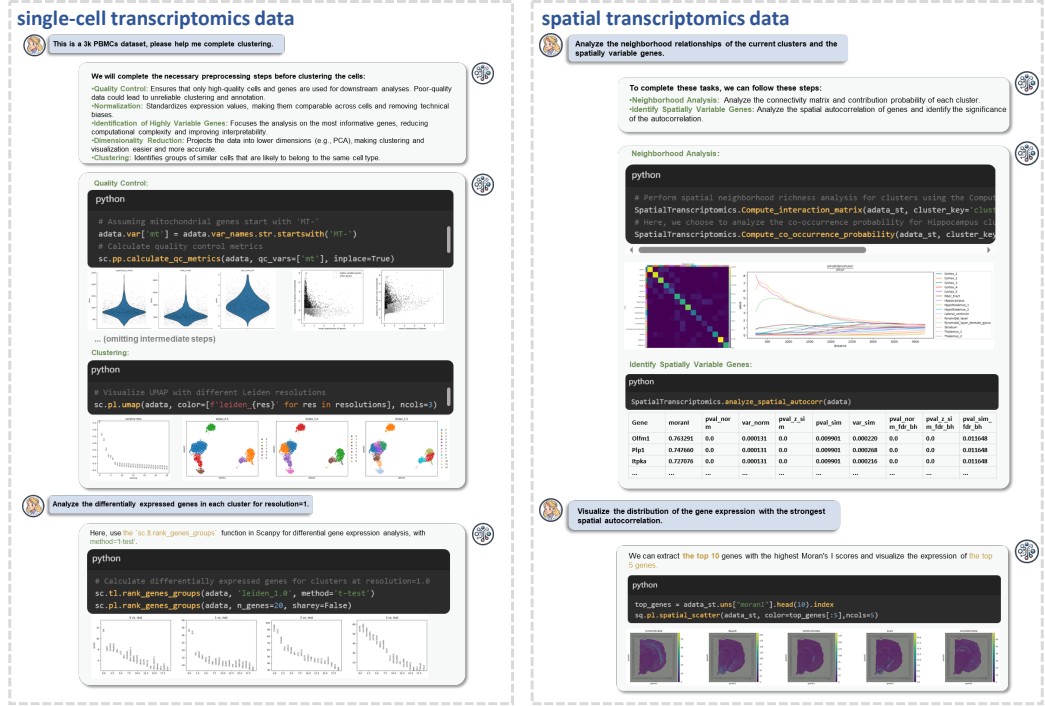

Figure 2: **Example of using CellAgent for single-cell RNA sequencing and spatial transcriptomics.** The user provides a dataset, CellAgent plans tasks, executes code, and generates results.

## 3 METHODS

The CellAgent framework automates and enhances single-cell and spatial transcriptomics analysis through a structured, multi-agent workflow, as illustrated in Figure 1. The system is built upon a hierarchical architecture that simulates a "deep-thinking" process, ensuring that complex analytical goals are decomposed and executed logically, and can be viewed as a hierarchical decision process over the space of analysis pipelines. First, a high-level Planner agent interprets a user's natural language request and formulates a coherent, multi-step analytical plan (Section 3.1). Next, each step of this plan is processed within a core operational loop where Executor and Evaluator agents collaborate to not only perform the analysis but also iteratively optimize the results for quality and accuracy (Section 3.2). In this view, the Planner is responsible for task decomposition, the Executor for instantiating concrete candidate pipelines, and the Evaluator for scoring and selecting among these candidates. This entire workflow is supported by a suite of essential modules including a comprehensive toolkit, a memory system, and a secure code sandbox that provide the necessary infrastructure for a robust and biologically sound analysis, which we detail in Section 3.3. This integrated approach enables CellAgent to autonomously produce high-quality, reproducible results, reliably completing tasks and interacting with the user via natural language, as demonstrated in Figure 2.

### 3.1 TASK PLANNING AND HIERARCHICAL DECOMPOSITION

The initial phase of the CellAgent workflow is managed by the Planner, an LLM-driven agent that functions as the system's high-level architect. Its primary responsibility is to translate a user's abstract analytical goal into a concrete, structured task plan. The Planner's system prompt ($p_{\text{sys}}^{p}$) is infused with expert knowledge of scRNA-seq and spatial transcriptomics workflows to ensure logical and sound task decomposition. This knowledge includes codified best-practices, such as the standard order of operations for common analyses (e.g., quality control must precede normalization), typical parameter ranges, and awareness of which downstream tasks require specific upstream pre-

processing steps. In practice, this stage determines the high-level structure of the analysis pipeline that subsequent agents will refine and execute.

Upon receiving a user's request—comprising the task description ($u_{\text{task}}$), the dataset ($D$), and any optional preferences ($u_{\text{req}}$), the Planner first inspects a summary of the dataset, $\psi(D)$, to ground its strategy in the data's specific characteristics. It then generates a sequence of discrete, ordered subtasks. Formally, this decomposition process by the Planner is defined as:

$$t_1, t_2, ..., t_n \leftarrow \mathcal{A}_p^{\text{LLM}}(p_{\text{sys}}^p, u_{\text{task}}, u_{\text{req}}, u_D, \psi(D)) \tag{1}$$

This hierarchical decomposition is crucial, as it transforms a broad objective into a manageable series of steps, setting a clear and logical foundation for the subsequent execution and optimization phase.

## 3.2 TASK EXECUTION VIA SELF-REFLECTIVE OPTIMIZATION MECHANISM

Following planning, each subtask is passed to the core operational engine of CellAgent, which executes the analysis and iteratively optimizes the outcome. This phase is a dynamic collaboration between the Executor and Evaluator agents, governed by a Self-Reflective Optimization mechanism.

The process begins with the **Executor** agent, which implements the subtask $t_i$. It consists of two components: a **Tool Selector** and a **Code Programmer**. The Tool Selector, $\mathcal{A}_t^{\text{LLM}}$, queries the available toolset $\mathcal{T}$ to identify the most appropriate tools, $\mathcal{T}_{t_i}$, for the current step:

$$\mathcal{T}_{t_i} \leftarrow \mathcal{A}_t^{\text{LLM}}(p_{\text{sys}}^t, u_{\text{req}}, \mathcal{T}, t_i) \tag{2}$$

Using the selected tools and their documentation, the Code Programmer, $\mathcal{A}_c^{\text{LLM}}$, generates executable Python code ($c_i$) and a textual analysis ($w_i$). This process is context-aware, leveraging a memory module $\mathcal{M}$ that contains the successful code from previous steps to ensure coherence:

$$(c_i, w_i) \leftarrow \mathcal{A}_c^{\text{LLM}}(p_{\text{sys}}^c, u_{\text{task}}, u_{\text{req}}, u_D, \psi(D), \mathcal{M}, t_i, \text{Doc}(\mathcal{T}_{t_i})) \tag{3}$$

In case of an execution error $\mathcal{E}(c_i)$, the Executor autonomously performs self-correction to produce a valid code version.

Upon successful generation of a result, the **Evaluator** agent, $\mathcal{A}_e^{\text{LLM}}$ assesses the outcome and provides natural language feedback if it deems revisions are necessary. Subsequently, when presented with multiple candidate solutions $\{c_i^j\}$, the Evaluator selects the optimal one $\bar{c}_i$, as the final output for the step:

$$\bar{c}_i = \mathcal{A}_e^{\text{LLM}}(p_{\text{sys}}^e, u_{\text{req}}, u_D, t_i, \{c_i^j\}), \quad j = 1, 2, ... \tag{4}$$

This evaluation is driven by the **Self-Reflective Optimization Mechanism**, a cornerstone of CellAgent. The process initiates by executing a suite of established algorithms for the given task, after which the Evaluator module, powered by GPT-4o, systematically assesses each method's output. The evaluation criteria are context-dependent and multifaceted, incorporating quantitative performance metrics (e.g., Accuracy Score for imputation and iLISI for batch correction), qualitative visual assessments guided by domain knowledge (e.g., for trajectory continuity and spatial domain coherence), and the synthesis of heterogeneous evidence (e.g., for cell type annotation). To ensure objectivity and mitigate biases from self-circularity, the Evaluator only receives anonymized outputs, task-specific metrics, and diagnostic plots for each candidate run, but never the underlying prompts or tool names produced by the Executor. CellAgent then selects the highest-scoring algorithm to produce the final result. A detailed breakdown of the optimization process for each task and the above fairness safeguards is available in the Appendix C.

## 3.3 KEY SUPPORTING MODULES

The planning and execution workflow of CellAgent is critically dependent on a set of modules that provide essential infrastructure for robustness, efficiency, and security.

**Memory Control** A key limitation of LLMs is their stateless nature. To overcome this, CellAgent integrates a dedicated memory module with a dual-architecture. This is based on the principle that its analytical subtasks are largely self-contained, depending on the final, validated outcome of

Table 1: **Comparison of multiple tasks**. cell type annotation, batch correction, trajectory inference, spatial domain identification, and spatial imputation.

| | | CellAgent | scGPT | SCSA | ScType | Celltypist | CellMaker |
|---|---|---|---|---|---|---|---|
| **Cell Type Annotation** | $\text{Average}_{\text{score}}$ | **0.85** | 0.77 | 0.47 | 0.59 | 0.75 | 0.58 |
| | | CellAgent | scVI | Liger | Harmony | Combat | ScGPT |
| **Batch Correction** | $\text{Batch}_{\text{correction}}$ | **0.67** | 0.66 | 0.65 | 0.60 | 0.56 | 0.50 |
| | $\text{Bio}_{\text{conservation}}$ | **0.66** | 0.65 | 0.64 | 0.61 | 0.58 | 0.62 |
| | $\text{Overall}_{\text{score}}$ | **0.67** | 0.66 | 0.65 | 0.60 | 0.57 | 0.57 |
| | | CellAgent | Slingshot | Scorpius | Page tree | Page | RaceID/StemID |
| **Trajectory Inference** | $\text{Cor}_{\text{dist}}$ | 0.48 | 0.48 | 0.43 | **0.49** | 0.44 | 0.21 |
| | $\text{F1}_{\text{branches}}$ | **0.83** | 0.71 | 0.79 | 0.48 | 0.33 | 0.22 |
| | $\text{Wcor}_{\text{features}}$ | 0.70 | **0.77** | 0.64 | 0.69 | 0.74 | 0.56 |
| | $\text{Edge}_{\text{flip}}$ | **0.88** | **0.88** | 0.83 | 0.61 | 0.14 | 0.20 |
| | $\text{Overall}_{\text{score}}$ | **0.50** | 0.47 | 0.44 | 0.40 | 0.07 | 0.06 |
| | | CellAgent | BayesSpace | DeppST | SEDR | spaGCN | stLearn |
| **Domain Identification** | ARI | **0.47** | 0.46 | **0.47** | 0.44 | 0.38 | 0.38 |
| | | CellAgent | Tangram | SpaGE | gimVI | nanoSpaRc | stPlus |
| **Spatial Imputation** | $\text{RANK}_{\text{PCC}}$ | **0.90** | 0.80 | 0.80 | 0.65 | 0.35 | 0.40 |
| | $\text{RANK}_{\text{RMSE}}$ | **0.90** | 0.80 | 0.75 | 0.65 | 0.35 | 0.45 |
| | $\text{RANK}_{\text{SSIM}}$ | **0.90** | 0.80 | 0.80 | 0.70 | 0.35 | 0.35 |
| | $\text{RANK}_{\text{JS}}$ | 0.80 | 0.60 | 0.55 | **0.85** | 0.60 | 0.40 |
| | $\text{Accuracy}_{\text{score}}$ | **0.88** | 0.75 | 0.73 | 0.71 | 0.41 | 0.40 |

the preceding one, not on intermediate trial-and-error processes. A **global memory** stores only the final code of each completed substep ($\mathcal{M} \leftarrow \{\bar{c}_1, \bar{c}_2, ...\}$), as illustrated in Fig. 1. The decision to store only code is deliberate: in bioinformatics analysis, code possesses high information entropy, concisely encapsulating complex data transformations. This strategy allows for the transfer of comprehensive contextual information with a minimal token footprint. In contrast, a **local memory** operates as a short-term workspace for the Executor within a single subtask. It captures the real-time execution trace, including all generated code snippets (both correct and incorrect), resulting error messages, and self-correction iterations. This allows the agent to learn from mistakes in real-time and avoid repeating errors before the local memory is discarded upon successful completion of the subtask. This discard-after-use strategy ensures that only the clear and successful analysis path is retained in the global memory, preventing interference from the intermediate trial-and-error history and thereby maintaining a concise and efficient context for subsequent steps.

**Tool Retrieval**   To provide comprehensive analytical capabilities, CellAgent is equipped with a diverse toolkit, sc-Omni ($\mathcal{T}$), for scRNA-seq and spatial transcriptomics analysis. This toolkit consolidates tools for over 15 distinct analytical tasks, ranging from preliminary analysis like quality control to advanced applications such as trajectory inference and spatial domain identification. All tools are registered within the framework, allowing the Tool Selector to query the available functions and provide a curated list of relevant options to the Code Programmer for each subtask. To ensure precise usage, each tool is implemented as a Python class or function, accompanied by standardized documentation via docstrings. This self-documenting design enables the programmer to dynamically retrieve detailed specifications, including parameters and return values, during runtime. This capability is essential for generating code that is both syntactically and semantically correct.

**Code Sandbox**   To ensure the security and reliability of code execution, CellAgent implements a Code Sandbox, which isolates the generated code during execution. This is implemented through Jupyter Notebook Conversion (`nbconvert`), where data loading and each step of the generated code are executed within a structured Jupyter Notebook environment. This approach effectively decouples the execution of single-cell data analysis from the core CellAgent framework, reducing potential risks associated with direct execution while improving system robustness. Additionally, by leveraging a notebook-based execution strategy, CellAgent enhances result management, facilitates workflow reproducibility, and ensures structured and traceable execution for all analysis tasks.

# 4 EXPERIMENTS

To comprehensively evaluate CellAgent, we conducted experiments on representative downstream tasks in both single-cell RNA sequencing and spatial transcriptomics. To ensure fair and comparable evaluation, we standardized preprocessing across all downstream tasks, controlling for both the number of cells and the set of gene expression features. For scRNA-seq, we assessed batch correction, cell type annotation, and trajectory inference; for spatial transcriptomics, we examined spatial domain identification and imputation (Table 1). We further benchmarked CellAgent against human experts and established frameworks in terms of both quality and computational efficiency across these tasks, and compared the performance of alternative agents on the same benchmarks (Appendix E.1 and E.2). In addition, we performed ablation studies to isolate the impact of key design choices. Specifically, we systematically evaluated CellAgent under different base models and analyzed the effect of enabling or disabling the memory optimization mechanism (Appendix E.3). These results provide deeper insights into the robustness, efficiency, and adaptability of CellAgent across diverse settings.

## 4.1 SINGLE-CELL RNA SEQUENCE DOWNSTREAM TASKS

Batch correction aims to adjust for technical variation across different batches of scRNA-seq data, ensuring that technical discrepancies do not distort biological signals (Haghverdi et al., 2018). To evaluate CellAgent, we compared it with other advanced methods across five datasets covering major tissues such as lung, pancreas, and perirhinal cortex, with varying numbers of cells, genes, and batch origins (Table 4 of Appendix). Ten metrics were used to assess batch effect removal and biological variance preservation, and an overall score was calculated as a weighted sum (see Appendix D.1). CellAgent outperformed other methods, achieving the highest average overall score (0.67), batch correction score (0.67), and bio-conservation score (0.66) across all datasets (Fig. 15 of Appendix). Visualization of human pancreas dataset further demonstrates that CellAgent effectively aligns cells of the same type across batches while preserving distinct separation of different cell types, particularly alpha, beta, and ductal cells (Fig. 16 and Fig. 17 of Appendix).

To evaluate CellAgent for cell-type annotation, we compared its predictions with other advanced methods across six datasets spanning diverse tissues, organs, and sequencing technologies, including Smart-seq2, 10X Chromium, and Drop-seq (Table 5 of Appendix). Using the Cell Ontology (CL), predicted annotations were categorized as "fully match," "partially match," or "mismatch," corresponding to consistency scores of 1, 0.5, and 0. The average consistency score was then calculated per dataset (see Appendix D.2). CellAgent achieved an average score of 85%. On the human PBMC dataset, it further refined existing annotations by subdividing CD8+ T cells into naive cytotoxic T cells and CD8+ T cells (Fig. 3a). This refinement is biologically meaningful because naive cytotoxic T cells represent a distinct subset of CD8+ T cells that have not yet encountered their specific antigen and remain undifferentiated into functional effector cytotoxic T cells (Lam et al., 2024). Additionally, CellAgent generated differential expression and marker gene visualizations, enabling intuitive interpretation of cluster identities and potential biological differences (Fig. 13).

Trajectory inference is intended to determine the pattern of a dynamic process experienced by cells and model cell state transitions (Trapnell et al., 2014). To assess the performance of CellAgent in trajectory inference, we compared it against five established trajectory inference methods across eight different datasets with gold-standard trajectory information (Saelens et al., 2019). We utilized four distinct types of metrics to evaluate different aspects of trajectories, and the overall score was calculated as a weighted average (details in Appendix D.3). CellAgent achieved the highest overall score of 0.50 (Table 1). Notably, in the "Aging HSC Kowalczyk" dataset, CellAgent successfully reconstructed the developmental trajectory from long-term hematopoietic stem cells (LT-HSC) to short-term hematopoietic stem cells (ST-HSC), and ultimately to multipotent progenitors (MPP) (Fig. 3b). This aligns with existing studies that describe a gradual transition from hematopoietic stem cells with high proliferative and renewal potential to those with a progressive loss of such potential, but a gain of differentiated features (Quesenberry et al., 2014). Additionally, CellAgent generated a heatmap of the top 20 differentially expressed genes along the trajectory (Fig. 14 in the Appendix), capturing the gradual upregulation of CD48 and MPO gene expression during the hematopoietic stem cell differentiation process, consistent with their known roles in immune cell maturation and myeloid differentiation (de Haan & Lazare, 2018). The finding supports the potential of CellAgent

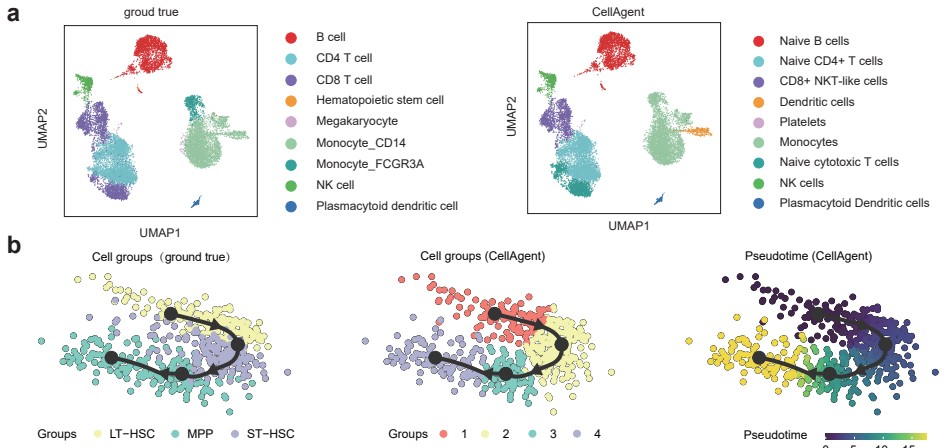

Figure 3: **Results of CellAgent in single-cell transcriptomics analysis. a,** UMAP visualization of cell type annotation from the human PBMC dataset, colored according to the cell types annotated in the original study and the cell types predicted by CellAgent. **b,** The trajectory UMAP plot of original cell grouping, reconstructed cell grouping based on trajectory milestones by CellAgent, and pseudotime for the Aging HSC Kowalczyk dataset, going from LT-HSC, ST-HSC to MPP.

in identifying key genes that play critical roles in cell fate determination. These results highlight the effectiveness of CellAgent as a powerful tool for both trajectory inference and identification of key regulatory genes.

## 4.2 SPATIAL TRANSCRIPTOMICS DOWNSTREAM TASKS

Spatial domain identification aims to delineate distinct tissue regions exhibiting spatial coherence based on spatial transcriptomic data (Hu et al., 2021). To evaluate CellAgent, we compared it with other advanced methods across 12 dorsolateral prefrontal cortex (DLPFC) slice datasets initially annotated by Maynard et al. (Maynard et al., 2021), where manual annotations delineated cortical layers (L1–L6) and white matter (WM). CellAgent achieved the highest average ARI score (0.47) across all datasets with a smaller interquartile range, indicating superior performance and robustness (Table 1). Visualization on slice 151673 (3639 spots, 33,538 genes) shows that CellAgent accurately captured the intricate spatial organization of the tissue, including the differentiation of cortical layers and the white matter regions, producing smoother, biologically consistent spatial domains (Fig. 4a).

Spatial transcriptomics imputation focuses on estimating the missing gene expression over the measured spots (Song et al., 2023). To evaluate CellAgent, we compared it with other advanced methods across seven datasets from multiple sequencing platforms, including 10X Visium, SlideseqV2, and seqFISH (Table 8 of Appendix). Performance was assessed using four metrics combined into an overall Accuracy Score (AS) (Li et al., 2022). CellAgent outperformed other methods across all evaluation metrics, including Pearson correlation coefficient (PCC), root mean square error (RMSE), structural similarity index (SSIM), and Jensen–Shannon divergence (JS). CellAgent achieved the highest average AS of 0.88, outperforming the suboptimal method, Tangram, by 17% (Table 1). It also demonstrated robust performance across diverse datasets. Specifically, on the mouse cortex dataset, the predicted expression of Igsf21, a marker gene in the mouse cortex (Tasic et al., 2018), closely aligned with the true expression patterns (Fig. 4b). Among the predicted genes, Ptgfrn, Zyx, and Rprm showed the highest Moran's I scores and were identified as top spatially variable genes. In particular, Rprm was predicted to be highly expressed in the L5/L6 layers of the cortex, consistent with its critical role in neuronal synaptic signaling and plasticity (Tasic et al., 2018).

## 4.3 HUMAN CENTERED EVALUATION

CellAgent enables natural language-driven analysis of scRNA-seq and spatial transcriptomics data. To systematically evaluate performance, we benchmarked CellAgent across eight major single-cell and spatial transcriptomics tasks on over 60 datasets (Tables 3 in Appendix), including preprocess-

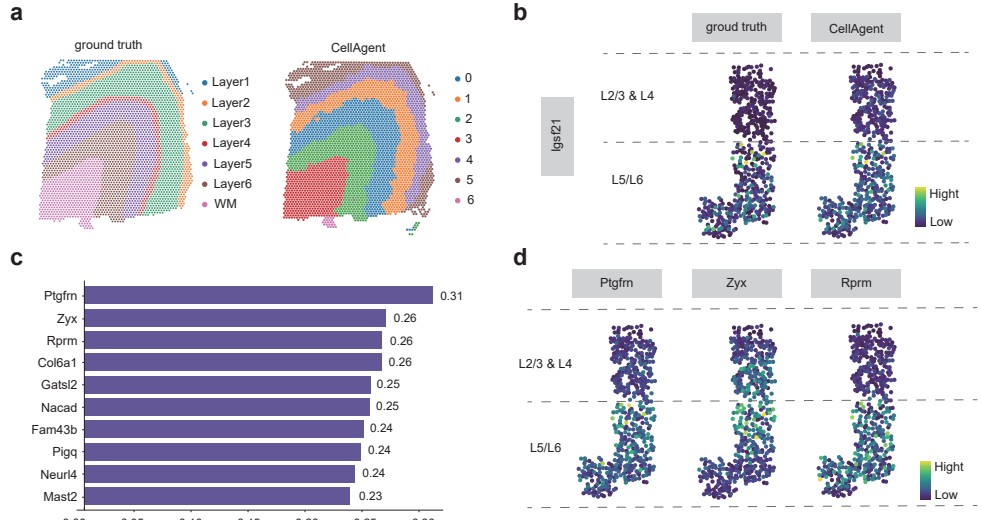

Figure 4: **Results of CellAgent in spatial transcriptomics analysis. a,** Visualization of the ground truth and CellAgent-identified spatial domains in slice 151673 of the DLPFC. **b,** The ground truth spatial distribution of the gene lgsf21 and the predicted distribution by CellAgent on the mouse cortex dataset. **c,** The top ten spatially variable genes were identified after imputation by CellAgent on the mouse cortex dataset. These genes were ranked based on Moran's I spatial autocorrelation scores. **d,** The spatial distribution of top three highly variable genes predicted by CellAgent.

ing, clustering, cell type annotation, batch correction, trajectory inference, spatial domain identification, spatial imputation, spatially variable gene identification, and neighborhood analysis. CellAgent was compared with human experts for efficiency and quality. On average, it completed tasks in 8 minutes versus 13 minutes for experts and achieved slightly higher quality scores (0.25 increase) (Fig. 5a and b). Compared to GPT-4, CellAgent@3, which supports up to three iterations of self-debugging, achieved an average execution success rate of 96%, far exceeding GPT-4's 24%. Usability assessment with 20 participants (10 experts, 10 novices) showed that 75% rated CellAgent as highly facilitative, surpassing GPT-4 (60%) and an online web server (30%) (Fig. 5c and d). In summary, compared with online servers, single-cell foundation models, task-specific methods, and GPT-4, CellAgent offers high-quality analysis, no manual coding, self-optimizing execution, and natural language interaction (Fig. 6 in Appendix), demonstrating its potential to advance scRNA-seq and spatial transcriptomics data analysis.

## 5 CONCLUSION

Single-cell RNA sequencing and spatial transcriptomics data analysis are essential for driving biological discoveries. However, conducting high-quality data analysis requires profound domain knowledge and proficient programming skills, and most existing tools lack support for natural language interaction and have limitations in delivering high-quality automated analysis of single-cell and spatial transcriptomics data. To address these limitations, we introduce CellAgent, an autonomous, LLM-driven framework that integrates natural language interactions, a multi-agent hierarchical decision-making system, and a self-reflective optimization mechanism to automate single-cell RNA sequencing data analysis. This approach simulates a "deep-thinking" workflow, systematically decomposing complex tasks, intelligently selecting tools, and iteratively refining results to achieve high-quality data analysis. Compared to single-cell foundation models, online Webserver, and other tools, CellAgent offers greater flexibility, broader task coverage, and an enhanced user experience through a natural language interactive, dialogue-based approach. Through extensive experiments on a diverse set of datasets, CellAgent has demonstrated both competitive execution efficiency and analytical quality, highlighting its potential for automated data analysis. In summary, CellAgent is a versatile and scalable tool that reduces the complexity and cost of single-cell RNA sequencing and spatial transcriptomics data analysis. The development of CellAgent will establish

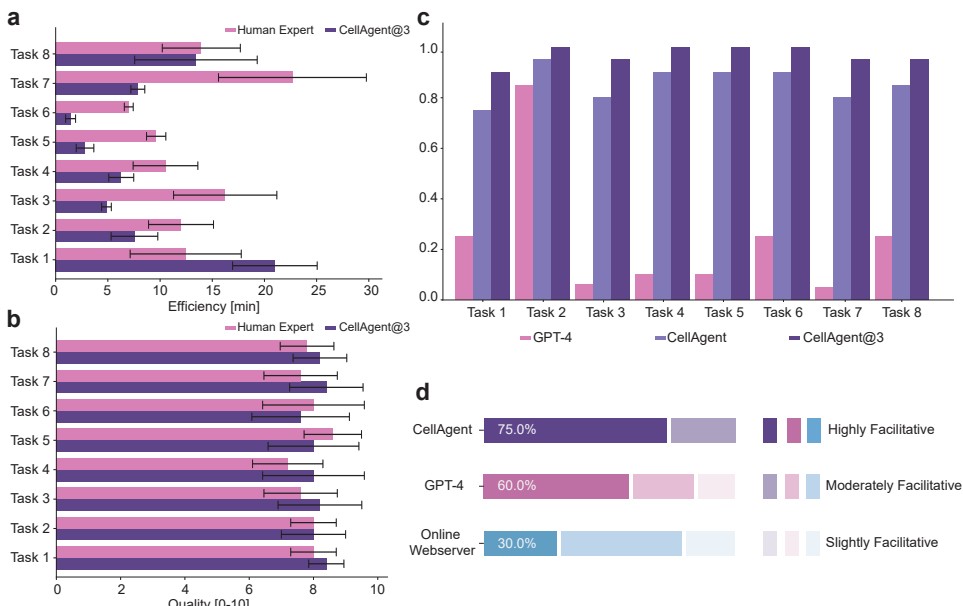

Figure 5: **Performance comparison. a,** Comparison of the efficiency of CellAgent and Human Expert on eight tasks involving scRNA-seq and spatial transcriptomics data analysis. Efficiency was assessed in minutes (Table 3 in Appendix). **b,** Comparison of the quality of CellAgent and Human Expert on eight tasks involving scRNA-seq and spatial transcriptomics data analysis. Quality was rated on a scale from 0 to 10, with evaluations conducted by evaluators (n=5). **c,** Comparison of the success rates of GPT-4, CellAgent, and CellAgent@3 (representing the best outcome after up to three attempts) across the 8 tasks. **d,** Assessment of the facilitation of CellAgent, GPT-4, and Online Webserver on scRNA-seq and data analysis, as evaluated by 20 participants.

a new paradigm in bioinformatics, broadening the role of generative AI in scientific discovery and potentially leading to novel insights into biological systems.

## ETHICS STATEMENT

The development and application of CellAgent were guided by principles of responsible AI and scientific integrity. All datasets utilized in this study for benchmarking and demonstration are publicly available and have been previously published, ensuring that no new private or sensitive patient data was collected. Our primary goal is to democratize single-cell and spatial transcriptomics analysis, making these powerful technologies more accessible to researchers who may lack extensive computational expertise. CellAgent is designed as an assistive tool to augment, not replace, human expertise.

Human participation was essential for evaluating the performance and usability of our framework. To benchmark the performance of CellAgent against current practices, we invited human experts to participate in assessments of efficiency and quality. All participating experts were provided with comprehensive information regarding the research objectives, evaluation procedures, and data usage, and gave their informed consent prior to their involvement. All collected performance data (e.g., time to complete tasks) and quality scores were fully anonymized. In the paper, results are presented solely in the form of aggregated statistics, ensuring that no personally identifiable information is disclosed. The participation of these experts was entirely voluntary, and they retained the right to withdraw at any stage without reason.

To ensure responsible deployment and mitigate potential risks, the CellAgent platform is intended for research purposes only. The authors declare no competing interests, and the research was conducted without any sponsorship that could have unduly influenced its outcomes.

## REPRODUCIBILITY STATEMENT

We have taken several steps to ensure the reproducibility of our work. We offer an online webpage accessible at `http://cell.agent4science.cn/`.

**Code and Service:** The complete sc-Omni toolkit and the source code for CellAgent are open-source and available at `https://github.com/23AIBox/cellagent`.

**Data:** All datasets used in our experiments are publicly available from their original sources, which are cited throughout the manuscript.

**Methodology:** The core logic of the CellAgent framework, including the roles and interactions of the Planner, Executor, and Evaluator agents, is described in detail in the Methods section. The specific prompts and interaction flows that govern the agents' behaviors are illustrated with examples in Appendix E, Figures 9-12. The LLM used for all agents is GPT-4o. While the model itself is proprietary, our detailed descriptions of the prompts and the agentic framework should enable researchers to replicate the system's behavior with similar state-of-the-art LLMs. The evaluation metrics for all downstream tasks are standard in the field and are explicitly defined with their corresponding formulas in Appendix D.

## 6 ACKNOWLEDGMENTS

This work is supported by the National Natural Science Foundation of China (Grant Nos. 62422605, 92370132, 62433016, 62072376, and 62553021), the National Key Research and Development Program of China (Grant No. 2024YFE0210900).

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

CONTENTS

# A    THE USE OF LLMS

In this work, LLMs were used solely in an assistive capacity. Their application was limited to aiding in code debugging and improving the language and clarity of the manuscript. All core scientific contributions, including the design of the framework, experimental results, and their interpretations, are the original work of the authors. Any suggestions provided by LLMs were carefully reviewed and verified. The authors take full responsibility for all content presented in this paper.

# B    DETAILS OF CELLAGENT

## B.1    FRAMEWORK DESIGN

CellAgent utilizes a collaborative framework of multiple expert-level large language models (LLMs) to achieve automated and high-quality analysis of single-cell RNA sequencing (scRNA-seq) and spatial transcriptomics (ST) data. Specifically, CellAgent leverages the expertise of three distinct roles: Planner, Executor, and Evaluator. This structured approach ensures that tasks are accurately decomposed and executed with minimal errors, providing reliable and professional results.

The Planner plays a crucial role in task decomposition. The system prompt explicitly defines the Planner's responsibilities and output format. It incorporates expert knowledge of scRNA-seq and spatial transcriptomics analysis to enhance reliability and minimize hallucinations. The user input includes task descriptions, user preferences, and data details. Based on this information, the Planner accurately interprets user intent and formulates a comprehensive analysis workflow. Subsequently, it decomposes the workflow into sub-tasks and outputs them in JSON format (Figure 9).

The role of the Executor is explicitly defined by the system prompt, which outlines its responsibilities, required input, and output format (Figure 10). The Executor integrates essential expert knowledge regarding code generation constraints and toolkit usage to ensure stability. The provided instructions contain a structured task and data description, along with documentation of available tools, thus ensuring consistency with the current step. Based on these instructions, the Executor generates and executes the Python code. In the event of errors, it automatically adjusts parameters using exception information to generate executable code (Figure 11).

The Evaluator assesses code execution outcomes based on a designed self-reflection mechanism (Figure 12). Specifically, it integrates multimodal large language models (GPT-4o) to facilitate the automated assessment of batch correction, trajectory inference, and spatial domain identification. Regarding cell-type annotation, the Evaluator aggregates prediction results from multiple tools to generate the final cell type labels. In the context of spatial transcriptomics imputation, the Evaluator assesses the predictive performance of different imputation methods against known gene expression data in the current dataset. Specifically, it compares the predicted expression values with the actual gene expression to determine the most effective approach, then the selected method is applied to infer the expression of unknown genes, facilitating subsequent analyses.

## B.2    EVALUATOR AND PARTICIPANT BACKGROUNDS

To ensure the scientific rigor and generalizability of the qualitative and quantitative assessments in this study (including Human Expert benchmarking, quality ratings, and usability evaluation), we recruited 20 researchers with varying levels of computational biology expertise. Ten of them formed the Expert Quality Assessment Panel, which was responsible for independently scoring (0–10) the outputs generated by CellAgent and the Human Experts. This panel consisted of master's and doctoral students with 2–3 years of experience in single-cell/spatial transcriptomics and relevant publications; among them, five simultaneously served as Human Experts, performing manual analyses to provide benchmark results, while the other five focused on comparative scoring. In addition, another ten graduate students who had just begun working with single-cell data (less than one year of experience) participated in the usability evaluation. Together with the aforementioned students, they formed the User Usability Assessment Group (n=20), which was tasked with evaluating the user-friendliness, facilitation, and interactive efficiency of CellAgent, the GPT-4 baseline, and the Online Webserver platform. This design enabled the capture of differential user perceptions and practical

utility across experience levels, thereby ensuring the comprehensiveness and scientific robustness of the evaluation outcomes.

## C SELF-REFLECTIVE OPTIMIZATION MECHANISM

### C.1 BATCH CORRECTION

CellAgent employs a Self-Reflective Optimization mechanism to invoke different batch correction methods (e.g., Harmony, scVI and Liger) and determine the most effective method. Since batch correction outcomes cannot be directly interpreted through textual descriptions, the Evaluator quantitatively assesses batch effect removal using batch correction metrics on batch labels, which are computed based on batch labels, including Graph Connectivity (Luecken et al., 2022), PCR Comparison (Luecken et al., 2022), $iLISI_{Graph}$ (Korsunsky et al., 2019), kBET (Luecken et al., 2022), and $ASW_{batch}$ (Büttner et al., 2019). These metrics comprehensively measure the degree of batch effect elimination across different correction methods. To further assess biological variance conservation after batch correction, the Evaluator, powered by GPT-4o, applies Leiden clustering on the corrected data and visualizes the cell distribution by UMAP, where cells are colored based on Leiden cluster labels and assigns a 0-10 score based on cluster quality. This score is weighted by a default coefficient of 0.1 and combined with batch correction metrics to compute the final performance score for each method. By default, the optimization process runs three iterations, each invoking a different batch correction algorithm, with CellAgent automatically selecting the highest-scoring method for final output. To mitigate potential biases in LLM-based evaluations, we mask algorithm names and other extraneous details during the assessment. This ensures that the Evaluator's judgment is based solely on visual and performance-related criteria rather than any inherent biases from the model's training data.

### C.2 CELL TYPE ANNOTATION

CellAgent integrates both database-based annotation tools, such as CellMarker 2.0 and ACT, and gene expression-based tools, such as CellTypist, to facilitate cell type annotation. Leveraging a self-reflective optimization mechanism, CellAgent autonomously selects and executes multiple annotation methods, generating predicted cell type labels. The Evaluator, powered by GPT-4, first collects these annotations together with cluster-level differentially expressed gene (DEG) information and then outputs the final cell type assignments to enhance annotation accuracy.

### C.3 TRAJECTORY INFERENCE

The evaluation of trajectory inference follows a similar approach to batch correction assessment. Considering the complexity of describing trajectory information through text alone, the Evaluator scores the visualization results of different trajectory inference algorithms and selects the highest-scoring outcome as the final output during CellAgent's iterative optimization process. Trajectory inference visualization consists of three concatenated images, each providing distinct insights: (1) a UMAP plot with trajectories colored by cell types; (2) a UMAP plot displaying cell type clusters optimized based on trajectory milestones; and (3) a UMAP plot colored by pseudotime. This multi-layered visualization strategy enhances the semantic richness of trajectory representations, improving interpretability. To further refine the evaluation, expert-curated knowledge on trajectory inference tasks is incorporated into the Evaluator's prompts, directing the Evaluator's attention to the biological validity of developmental trajectories among different cell types in the plots and the continuity of the inferred trajectories. Additionally, to ensure fairness and eliminate bias, algorithm names are masked during evaluation, allowing the selection process to be driven solely by objective performance metrics.

### C.4 SPATIAL DOMAIN IDENTIFICATION

CellAgent applies multiple spatial domain identification algorithms, such as DeepST, stLearn, and SEDR, autonomously selecting and executing the most suitable method to generate spatial domain identification results. The Evaluator, powered by GPT-4o, analyzes spatial domain images predicted by different algorithms, which are generated by spatial visualizations based on predicted clustering

assignments, alongside original pathological images. Utilizing domain-specific knowledge of spatial partitioning, the Evaluator assesses the clustering quality within identified spatial domains and the smoothness of spatial domain boundaries, assigning scores to the spatial domain images produced by each algorithm on a 0 to 10 scale. The highest-scoring algorithm is then selected, and its corresponding spatial domain image is designated as the final result for spatial domain identification.

### C.5 IMPUTATION FOR SPATIAL TRANSCRIPTOMICS

CellAgent initially selects a subset of genes with known expression, which serves as both the training and validation set. These genes, known as highly variable genes (HVGs), exhibit high expression variability in both the spatial transcriptomics dataset and the reference single-cell RNA sequencing data. CellAgent then employs a range of imputation methods, such as Tangram, gimVI, and SpaGE. To evaluate performance, the Evaluator compares the predicted expression values on the validation set genes with their actual expression values. Different algorithms are then assessed based on their performance on this dataset, calculating the Accuracy Score (AS). The highest-scoring algorithm is identified as the most promising, and subsequently, CellAgent invokes this method to predict the expression of unknown genes.

### C.6 MECHANISMS FOR MITIGATING EVALUATION BIAS AND ENSURING ROBUSTNESS

The Self-Reflective Optimization Mechanism in CellAgent operates by running multiple candidate pipelines for each analysis step, scoring their outputs, and iteratively refining tool choices and hyperparameters. The task-specific scoring functions and optimization loop are described in detail in the main text; here we focus on how this mechanism is implemented to minimize evaluation bias and ensure robustness.

Because the Evaluator is itself an LLM that both interprets results and selects preferred tools, there is a natural risk of evaluation bias and self-circularity—for example, favoring methods whose names are frequent in its pre-training data or implicitly justifying its own earlier decisions. To counteract this, we adopt three complementary safeguards. First, all algorithm identifiers are masked before evaluation: method names, implementation-specific strings, and hyperparameter presets are replaced by generic labels ("Method A/B/C"), preventing the Evaluator from exploiting popularity or brand recognition of specific tools. Second, execution and evaluation memories are strictly separated. The Executor has access to full code, stack traces, and detailed logs for self-debugging, whereas the Evaluator's context is restricted to (1) a dataset-level summary, (2) anonymized candidate identifiers, (3) the corresponding task-specific metrics, and (4) rendered diagnostic plots, so it cannot simply rationalize the underlying code or prompts. Third, we empirically assess cross-model robustness by instantiating CellAgent with GPT-4o, Qwen3-VL-235B, and Qwen3-Omni-30B. Across five core downstream tasks (batch correction, cell-type annotation, trajectory inference, spatial domain identification, and spatial imputation), the aggregate scores vary only modestly and, crucially, the ranking of preferred tools remains stable, supporting the claim that the Evaluator's decisions are driven primarily by task-specific metrics and diagnostics rather than idiosyncratic biases of a particular backbone.

## D EVALUATION METRIC CALCULATIONS

### D.1 BATCH CORRECTION

We adopted the evaluation metric calculations outlined by Luecken et al. (2022). in their benchmark study. Each metric is described below.

**NMI.** To quantify the concurrence between the cell type labels based on ground truth and the cluster labels obtained from integrated cell embeddings, we computed the normalized mutual information (NMI) score. NMI scores of 0 or 1 correspond to uncorrelated clustering or a perfect match, respectively.

**ARI.** The Rand index compares the overlap of two clusterings; it considers both correct clustering overlaps while also counting correct disagreements between two clusterings. Similar to NMI, we

compared the cell-type labels with the NMI-optimized clustering computed on the integrated dataset. An ARI of 0 or 1 corresponds to random labeling or a perfect match, respectively.

**ASW.** The silhouette width assesses the relationship between a cell's within-cluster distances and its distances to the closest cluster boundaries. By averaging the silhouette widths of all cells, we calculate the average silhouette width (ASW) score. This score ranges from -1 to 1, where a score of 1 indicates well-separated clusters, while scores from -1 to 0 suggest overlapping clusters and misclassification. For evaluating cell type clustering, we compute the ASW score based on cell type labels, represented as $ASW_{cell}$. To obtain this score, we utilize the following formula:

$$ASW_{cell} = (ASW_C + 1)/2$$

Here, C represents the cell types.

Regarding batch mixing evaluation, we calculate the ASW score considering batch labels and adjust it by subtracting 1. This score is denoted as $ASW_{batch}$. The calculation is as follows:

$$ASW_{batch} = 1 - |ASW_B|$$

Both $ASW_{cell}$ and $ASW_{batch}$ have values between 0 and 1. Higher scores indicate better cell-type clustering or batch-mixing performance.

**Graph connectivity.** The graph connectivity metric quantifies the average proportion of cells within each cell type connected through a kNN(k-nearest neighbors) graph. For every cell identity $c$ in the set *C*, we compute the size of the largest connected component using kNN among cells exclusively belonging to identity $c$. This value is divided by the total number of cells with identity c to obtain a normalized measure. The *GraphConn*(Graph connectivity) score is then reported as the average across all cell types:

$$GraphConn = \frac{1}{|C|} \sum_{c \in C} \frac{\left| LCC \left( G_c^{kNN} \right) \right|}{N_c}$$

Here, *LCC* represents the largest connected component, and *N* denotes the number of cells of each cell type.

**Principal component regression.** Principal component regression, derived from PCA, has previously been used to quantify batch effect removal. Briefly, the $R^2$ was calculated from a linear regression of the covariate of interest (for example, the batch variable *B*) onto each principal component. The variance contribution of the batch effect per principal component was then calculated as the product of the variance explained by the *i*th principal component(PC) and the corresponding $R^2 (PC_i \mid B)$. The sum across all variance contributions by the batch effects in all principal components gives the total variance explained by the batch variable as follows:

$$Var(C \mid B) = \sum_{i=1}^{G} Var(C|PC_i) \times R^2 (PC_i|B)$$

where $Var(C|PC_i)$ is the variance of the data matrix C explained by the *i*th principal component.

**kBET.** The kBET algorithm (v.0.99.6, release 4c9dafa) determines whether the label composition of a k nearest neighborhood of a cell is similar to the expected (global) label composition. The test is repeated for a random subset of cells, and the results are summarized as a rejection rate over all tested neighborhoods. Here we use the kBET(extended by Luecken et al. (2022)) to compare integration results on kNN graphs irrespective of the integration output format. This score ranges from 0 to 1, where a score of 1 indicates a successful removal of batch effects, while a score of 0 suggests a poor performance.

**Graph LISI.** The LISI, a diversity score, was proposed to assess both batch mixing(iLISI) and cell-type separation(cLISI). LISI scores are computed from neighborhood lists per node from integrated kNN graphs. Specifically, the inverse Simpson's index determines the number of cells that can be drawn from a neighbor list before one batch is observed twice. Thus, LISI scores range from 1 to N, where N is the total number of batches in the dataset. Here we use the LISI scores extended by Luecken et al. (2022) as $cLISI_{Graph}$, where a 0 value corresponds to low cell type separation. $iLISI_{Graph}$, where a 0 value corresponds to low batch integration.

**Aggregated metrics.** The aggregated metric $AvgBatch$ computes the average of batch effect removal metrics:

$$AvgBatch = (ASW_{batch} + iLISI_{Graph} + GraphConn + kBET + PCR\ comparison)/5$$

The aggregated metric $AvgBio$ computes the average of batch effect removal metrics:

$$AvgBio = (Isolated\ labels + NMI + ARI + ASW_{cell} + cLISI_{Graph})/5$$

The $Overall$ score, derived from averaging these metrics, serves as the final performance indicator:

$$Overall = AvgBatch \times 0.4 + AvgBio \times 0.6$$

### D.2 CELL TYPE ANNOTATION

To assess the accuracy of cell type annotations and facilitate comparisons across various methodologies, we employ an average scoring metric (Hou & Ji, 2024). For predicted and actual labels, "fully match" is awarded when they directly align in terms of annotation terms or CL cell ontology names. "partially match" occurs when labels share a general cell type category but differ in specific annotations or ontology names. Conversely, "mismatch" is declared when there is a discrepancy in broad cell type categories, annotations, or ontology names. We assign consistency scores of 1, 0.5, and 0 for "fully match," "partially match," and "mismatch" respectively (Hou & Ji, 2024). These scores are then averaged across all cell types within each dataset. This average score serves as a quantitative measure of annotation accuracy, enabling robust comparisons between different annotation methods and datasets. This systematic scoring approach ensures that researchers can objectively assess the precision of cell type classifications and refine their analytical methods or data interpretation as needed.

$$Average_{score} = 1 \times \text{"fully match"} + 0.5 \times \text{"partially match"} + 0 \times \text{"mismatch"}$$

### D.3 TRAJECTORY INFERENCE

We adopted the evaluation metric calculations outlined by Saelens et al (Saelens et al., 2019). in their benchmark study. Each metric is described below.

**Correlation between geodesic distances.** $cor_{dist}$ measures the correlation between geodesic distances. When the position of a cell is the same in both the reference and the prediction, its relative distances to all other cells in the trajectory should also be the same. This observation is the basis for the $cor_{dist}$ metric.The geodesic distance is the distance a cell has to go through the trajectory space to get from one position to another. The $cor_{dist}$ score ranges from 0 to 1. A higher score indicates a stronger correlation between geodesic distances.

**Edit distance between two trajectory topologies.** We used the $edgeflip$ score to assess the similarity in topology between two trajectories, regardless of where the cells were positioned. The $edgeflip$ score is defined as the minimal number of edges that need to be added or removed to transform one network into the other, divided by the total number of edges in both networks. This problem is equivalent to the maximum common edge subgraph problem, providing a quantitative measure of structural differences between trajectory topologies.

**The quality of clustering within the trajectory.** $F1_{branches}$ is used to evaluate the clustering quality of cells within the trajectory. The simplest way to calculate the similarity between the cellular positions of two topologies is by mapping each cell to its closest branch. These clusters of cells can then be compared using one of the many external cluster evaluation measures. The *F1* score maps two cluster sets by using their shared members based on the *Jaccard* similarity. It then calculates the *Recovery* as the average maximal *Jaccard* for every cluster in the first set of clusters. Conversely, the *Relevance* is calculated based on the average maximal similarity in the second set of clusters (in our case, the prediction). Both the *Recovery* and *Relevance* are then given equal weight in a harmonic

mean (*F1*). Formally, if $C$ and $C'$ are two cell clusters:

$$Jaccard\,(c, c') = \frac{|c \cap c'|}{|c \cup c'|}$$

$$Recovery = \frac{1}{|C|} \sum_{c \in C} \max_{c' \in C'} Jaccard\,(c, c')$$

$$Relevance = \frac{1}{|C'|} \sum_{c' \in C'} \max_{c \in C} Jaccard(c, c')$$

$$F1 = \frac{2}{\frac{1}{Recovery} + \frac{1}{Relevance}}$$

To calculate $F1_{branches}$, cells are mapped to the closest edge within the trajectory, ensuring that the branch assignments reflect the structural similarity between the two topologies.

**The accuracy of dynamical differentially expressed genes.** $wcor_{features}$ measures the accuracy of dynamically differentially expressed genes. The main advantage of studying cellular dynamic processes using single-cell data is that the dynamics of gene expression can be analyzed across the whole transcriptome. This enables the construction of models such as dynamic regulatory networks and gene expression modules. Such analyses rely on a sufficiently accurate cellular ordering to identify dynamically differentially expressed genes. To prioritize the most important differentially expressed features, we implemented $wcor_{features}$, which weights the correlation using the feature importance scores in the reference. This metric ranges from 0 to 1, where a higher score indicates a stronger correlation, reflecting the ability of the method to accurately capture the dynamics of gene expression.

**Aggregated metric.** To provide the overall metric, we therefore chose a metric which weighs every aspect of the trajectory equally: $cor_{dist}$ accessing ordering; $F1_{branches}$ accessing branch assignment; $edgeflip$ accessing topology; $wcor_{features}$ accessing the accuracy of differentially expressed features. The final overall score for a method was thus defined as:

$$Overall = \sqrt[4]{cor_{dist} \times edgeflip \times F1_{branches} \times wcor_{features}}$$

### D.4 SPATIAL DOMAIN IDENTIFICATION

To assess the accuracy of spatial domain identification, we use the adjusted rand index (ARI) as an evaluation metric. ARI is a measure of the consistency between clustering results and reference labels, ranging from -1 to 1. A value of 1 indicates perfect agreement, 0 indicates random agreement, and negative values indicate disagreement. We use the following formula to calculate the ARI:

$$ARI = \frac{\sum_{ij} \binom{n_{ij}}{2} - \left[ \sum_i \binom{a_i}{2} \sum_j \binom{b_j}{2} \right] / \binom{n}{2}}{\frac{1}{2} \left[ \sum_i \binom{a_i}{2} + \sum_j \binom{b_j}{2} \right] - \left[ \sum_i \binom{a_i}{2} \sum_j \binom{b_j}{2} \right] / \binom{n}{2}}$$

where $n$ is the total number of samples, $n_{ij}$ is the number of samples that belong to both reference cluster $i$ and predicted cluster $j$, $a_i$ is the number of samples in reference cluster $i$, and $b_j$ is the number of samples in predicted cluster $j$. By calculating the ARI for each dataset, we can quantify the accuracy of spatial domain identification.

### D.5 IMPUTATION FOR SPATIAL TRANSCRIPTOMICS

We adopted the evaluation metric calculations outlined by Li et al. (Li et al., 2022). in their benchmark study. Each metric is described below.

**PCC.** Quantifies the correlation between the ground truth and predicted spatial expression vectors for each gene. A higher Pearson correlation coefficient (PCC) indicates better prediction accuracy:

$$PCC = \frac{E\left[ (\tilde{x}_i - \tilde{u}_i)(x_i - u_i) \right]}{\tilde{\sigma}_i \sigma_i}$$

where $x_i$ and $\tilde{x}_i$ are the spatial expression vectors, $u_i$ and $\tilde{u}_i$ are their means, and $\sigma_i$ and $\tilde{\sigma}_i$ are their standard deviations.

**SSIM.** To evaluate the similarity of scaled gene expressions, with values in [0, 1]. Expressions are scaled as:

$$x'_{ij} = \frac{x_{ij}}{\max(x_{i1}, \ldots, x_{iM})}$$

Structural Similarity Index Measure (SSIM) is calculated as:

$$SSIM = \frac{(2\tilde{u}_i u_i + C_1)(2\text{cov}(x'_i, \tilde{x}'_i) + C_2)}{(\tilde{u}_i^2 + u_i^2 + C_1)(\tilde{\sigma}_i^2 + \sigma_i^2 + C_2)}$$

where $\text{cov}(x'_i, \tilde{x}'_i)$ is the covariance, and $C_1 = 0.01, C_2 = 0.03$ are stability constants. A higher SSIM indicates better prediction accuracy.

**RMSE.** Root Mean Squared Error (RMSE) calculates the error between the $z$-scores of the ground truth and predicted spatial expression:

$$RMSE = \sqrt{\frac{1}{M} \sum_{j=1}^{M} (z_{ij} - \tilde{z}_{ij})^2}$$

where $z_{ij}$ and $\tilde{z}_{ij}$ are $z$-scores. Lower RMSE indicates better prediction accuracy.

**JS.** Jensen-Shannon (JS) divergence measures differences in spatial distribution probabilities. The probability is computed as:

$$P_{ij} = \frac{x_{ij}}{\sum_{j=1}^{M} x_{ij}}$$

The JS divergence is calculated as follows:

$$JS = \frac{1}{2} KL \left( P_i \| \frac{\tilde{P}_i + P_i}{2} \right) + \frac{1}{2} KL \left( \tilde{P}_i \| \frac{\tilde{P}_i + P_i}{2} \right)$$

where the Kullback-Leibler divergence $KL(a\|b)$ is defined as:

$$KL(a\|b) = \sum_{j=1}^{M} a_j \log \left( \frac{a_j}{b_j} \right)$$

A lower JS value indicates better prediction accuracy.

**Aggregated metrics.** AS combines four metrics: PCC, SSIM, RMSE, and JS to evaluate overall performance. The methods are ranked as follows: ascending order for PCC and SSIM, where higher values indicate better performance, and descending order for RMSE and JS, where lower values indicate better performance. The method with the highest RMSE/JS value is assigned $\text{RANK}_{\text{RMSE/JS}} = \frac{1}{N}$, while the method with the lowest RMSE/JS value is assigned $\text{RANK}_{\text{RMSE/JS}} = 1$. The method with the highest SSIM/JS value is assigned RANKSSIM/JS $= 1$, while the method with the lowest SSIM/JS value is assigned RANKSSIM/JS $= \frac{1}{N}$. Finally, we calculated the average value of $\text{RANK}_{\text{PCC}}$, $\text{RANK}_{\text{SSIM}}$, $\text{RANK}_{\text{RMSE}}$, and $\text{RANK}_{\text{JS}}$ to obtain the AS value of each integration method, as follows:

$$AS = \frac{1}{4}(\text{RANK}_{\text{PCC}} + \text{RANK}_{\text{SSIM}} + \text{RANK}_{\text{RMSE}} + \text{RANK}_{\text{JS}})$$

A higher AS value indicates better overall performance.

# E SUPPLEMENTARY FIGURES

## E.1 PERFORMANCE COMPARISON OF CELLAGENT WITH OTHER METHODS

| Methods | CellAgent | GPT-4 | Online Webserver | Single-cell FMs | Task-specific Tools |
|---|---|---|---|---|---|
| High-quality Analysis | ✓ | ✗ | ✓ | ✓ | ✓ |
| No manual Coding | ✓ | ✓ | ✓ | ✗ | ✗ |
| Self-optimizing Execution | ✓ | ✗ | ✗ | ✗ | ✗ |
| Natural language Interaction | ✓ | ✓ | ✗ | ✗ | ✗ |

Figure 6: **Performance comparison of CellAgent with other methods.** Comprehensive comparison of CellAgent, GPT-4, online web servers, single-cell foundation models (single-cell FMs), and task-specific tools.

E.2   PERFORMANCE COMPARISON OF CELLAGENT WITH OTHER LLM-DRIVEN AGENTS

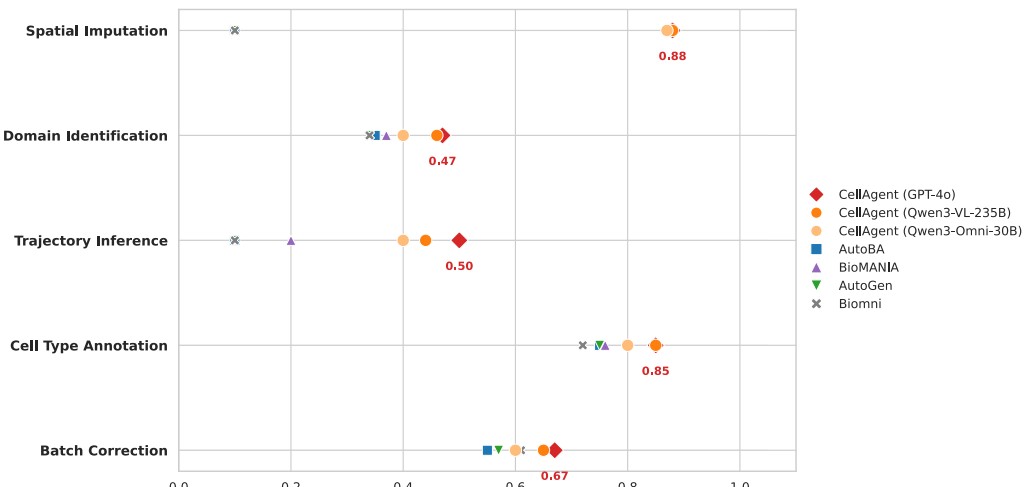

Figure 7: **Performance comparison of CellAgent with other single-cell analysis agents.** We evaluated the performance across five downstream tasks: Spatial Imputation, Domain Identification, Trajectory Inference, Cell Type Annotation, and Batch Correction. The x-axis represents the performance metric score (see Appendix D.1). CellAgent, particularly the version powered by GPT-4o (indicated by red diamonds), consistently outperforms baseline methods.

### E.3 MEMORY MECHANISM OPTIMIZATION ACROSS DIFFERENT BASE MODELS IN CELLAGENT

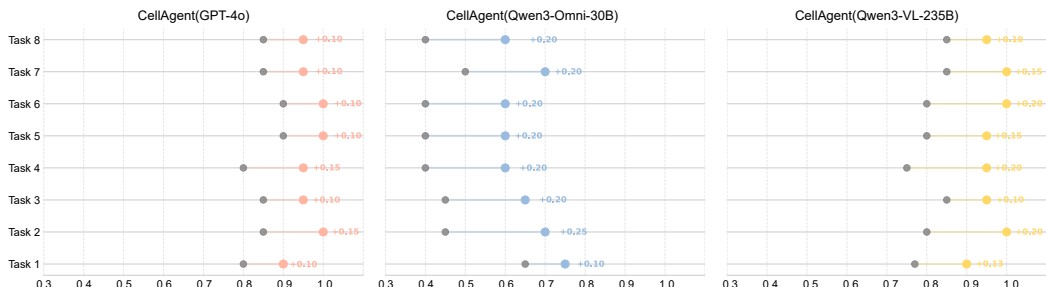

Figure 8: **Memory Mechanism Optimization across Different Base Models in CellAgent.** We evaluate the performance gain in task execution success rate provided by the memory optimization mechanism across eight analysis tasks (detailed in Table 3) and three distinct base models. The gray dots represent the baseline success rate of the model without memory optimization, while the colored markers and annotated numerical values display the performance gain contributed by enabling the memory optimization mechanism.

### E.4 DETAILS OF THE PLANNER

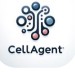
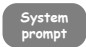

**Planner**

**# Role**
You are an expert in the field of biological analysis. You need to decompose a required task (such as debatch, cluster, et al.) into several sub-steps. Be sure to consider the requested steps as the final sub-task.

**# Expert Knowledge**
The set of all possible sub-steps (specific task components) contains: {Quality Control, Normalization, Identification of Highly Variable Genes, Dimensionality Reduction, Batch Effect Correction, Clustering, Cell Type Annotation, Cell-to-Cell Communication Analysis, Trajectory inference}.

The following attempts is for reference:
1. Batch Effect correction is only necessary for data with batch effects (there are batch-related labels such as "batch" or "sample" in the data). Note that Batch Effect Correction is also known as "debatch".
2. Batch Effect Correction should be ...
 ...

**# Output format instruction**
1. Use ordered lists in JSON format to provide overall steps, along with appropriate explanations.
...

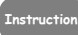

Task Description: Using the given Adata data to do cell type annotation.
User Requirements: No need for steps of Quality Control and Normalization.
Data Description: hum_lung_pp.h5ad
Data Format: AnnData object with n_obs × n_vars = 32472 × 32566
  obs: 'dataset', 'location', 'nGene', 'nUMI', 'patientGroup', 'percent.mito', 'protocol', 'batch', 'donor'
  var: 'gene_ids'
  layers: 'counts'

Please provide sub-steps to complete the task. Note that if there are user-provided requirements, they must be satisfied.

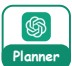

```json
[
    {
        "step": 1,
        "name": "Identification of Highly Variable Genes",
        "description": "Identify and select genes that exhibit high variability across cells,
which are often informative for distinguishing cell types and states."
    },
    {
        "step": 2,
        "name": "Batch Effect Correction",
        "description": "Correct for batch effects in the data to ensure that variations due to
different batches do not confound the biological signal."
    },
    {
        "step": 3,
        "name": "Dimensionality Reduction",
        "description": "Reduce the dimensionality of the dataset to capture the most significant
features of the data in a lower-dimensional space, often using methods like PCA, t-SNE, or UMAP."
    },
    {
        "step": 4,
        "name": "Clustering",
        "description": "Group cells into clusters based on their gene expression profiles to
identify distinct cell populations."
    },
    {
        "step": 5,
        "name": "Cell Type Annotation",
        "description": "Assign cell type labels to the identified clusters based on known marker
genes or reference datasets."
    }
]
```

Figure 9: **Details of the Planner.** The system prompt defines the Planner's role and output format while integrating domain-specific knowledge of single-cell analysis, including key sub-steps and task-specific constraints, which helps mitigate hallucinations, ensuring accurate and professional task decomposition. Structured instructions refine user input into itemized content, enhancing workflow efficiency. Additionally, the instructions specify a parsed data format, enabling precise interpretation of input data. Adhering to these guidelines, the Planner outputs decomposed sub-tasks in JSON format.

## E.5 DETAILS OF THE EXECUTOR

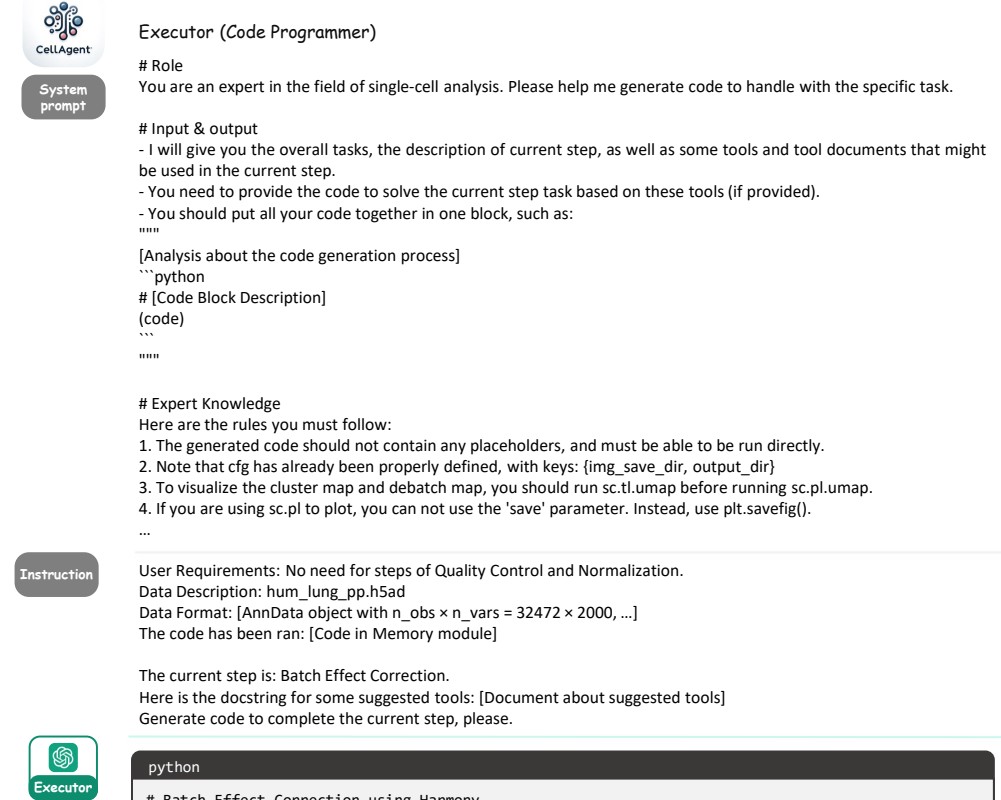

**Executor (Code Programmer)**

# Role
You are an expert in the field of single-cell analysis. Please help me generate code to handle with the specific task.

# Input & output
- I will give you the overall tasks, the description of current step, as well as some tools and tool documents that might be used in the current step.
- You need to provide the code to solve the current step task based on these tools (if provided).
- You should put all your code together in one block, such as:
"""
[Analysis about the code generation process]
```python
# [Code Block Description]
(code)
```
"""

# Expert Knowledge
Here are the rules you must follow:
1. The generated code should not contain any placeholders, and must be able to be run directly.
2. Note that cfg has already been properly defined, with keys: {img_save_dir, output_dir}
3. To visualize the cluster map and debatch map, you should run sc.tl.umap before running sc.pl.umap.
4. If you are using sc.pl to plot, you can not use the 'save' parameter. Instead, use plt.savefig().
...

**Instruction**

User Requirements: No need for steps of Quality Control and Normalization.
Data Description: hum_lung_pp.h5ad
Data Format: [AnnData object with n_obs × n_vars = 32472 × 2000, ...]
The code has been ran: [Code in Memory module]

The current step is: Batch Effect Correction.
Here is the docstring for some suggested tools: [Document about suggested tools]
Generate code to complete the current step, please.

```python
# Batch Effect Correction using Harmony
import scanpy as sc
import matplotlib.pyplot as plt
from tools.debatcher import Debatcher

# Instantiate the Debatcher class
debatcher = Debatcher(adata, batch='batch', cfg)

# Apply Harmony batch correction
adata = debatcher.harmony()

# Update the PCA with Harmony corrected data
adata.obsm['X_pca'] = adata.obsm['X_pca_harmony']
# Compute the neighborhood graph using the corrected PCA
sc.pp.neighbors(adata, n_neighbors=10, use_rep='X_pca')
# Perform Leiden clustering
sc.tl.leiden(adata, resolution=1)
# Run UMAP for visualization
sc.tl.umap(adata)

# Plot the UMAP to visualize the batch effect correction
sc.pl.umap(adata, color=['batch', 'leiden'], show=False)
plt.savefig(f"{cfg['img_save_dir']}/umap_plot_corrected.png")
```

Figure 10: **Details of the Executor**. The system prompt explicitly defines the role, required input, and output format of the Executor while integrating essential expert knowledge. This expert knowledge includes various constraints on code generation and the use of specific tools to enhance the stability of the generated code. The instructions provide a structured description of the task and data. Additionally, the documentation of available tools for the current step is included, ensuring consistency with the tool list retrieved by the Executor. Following these instructions, the Executor generates and outputs the Python code.

## E.6 DETAILS OF AUTOMATIC CODE EXCEPTION CORRECTION BY THE EXECUTOR

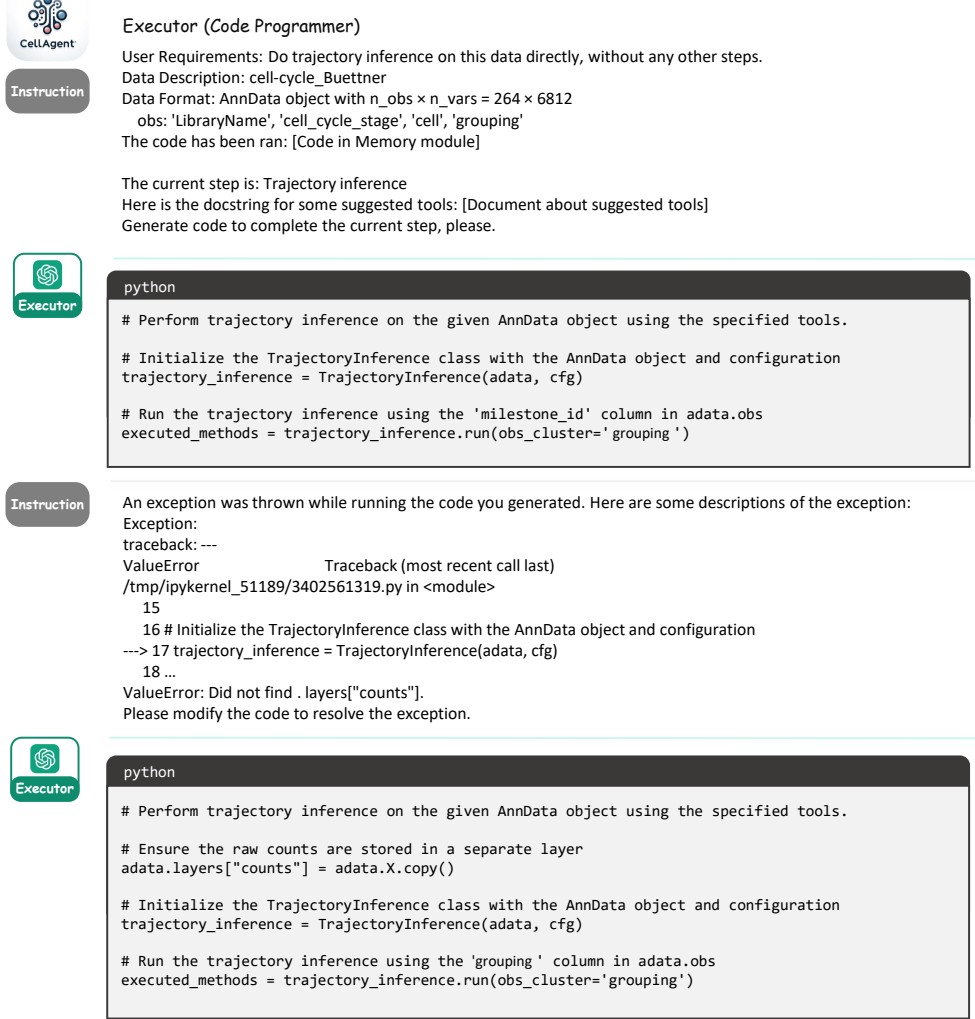

Figure 11: **Details of automatic code exception correction by the executor**. The Executor first outputs code that can not be successfully executed. This exception is then prompted to the Executor. Based on the exception information, the Executor automatically adjusts parameters to generate code that can be executed correctly.

## E.7 DETAILS OF THE EVALUATION PROCESS

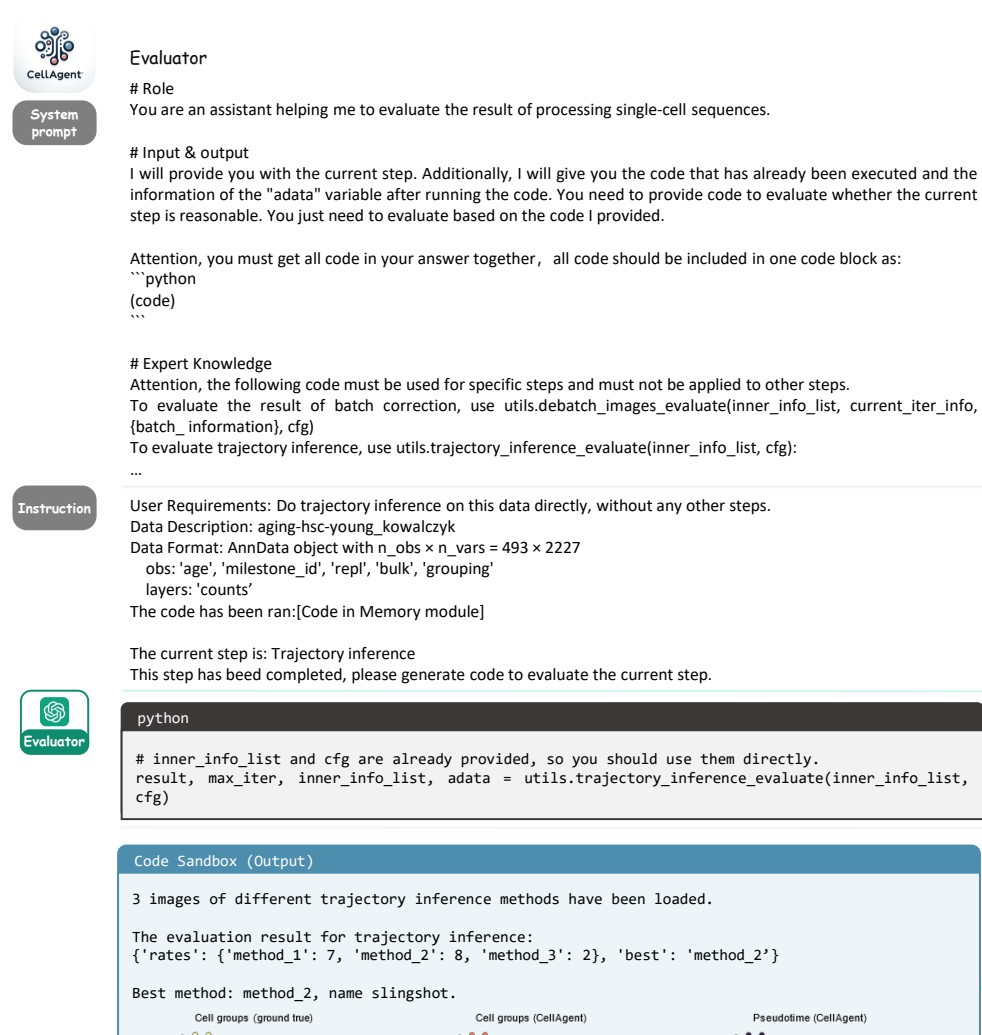

Figure 12: **Details of the Evaluation Process**. The system prompt for the Evaluator specifies its role, required inputs and outputs, and the essential expert knowledge necessary to enable the effective utilization of the self-reflective optimization mechanism. At each step, the Evaluator receives a structured description of the current task and data, which it subsequently analyzes to assess the corresponding results.

## E.8 RESULTS ON CELL TYPE ANNOTATION THROUGH DIALOGUE

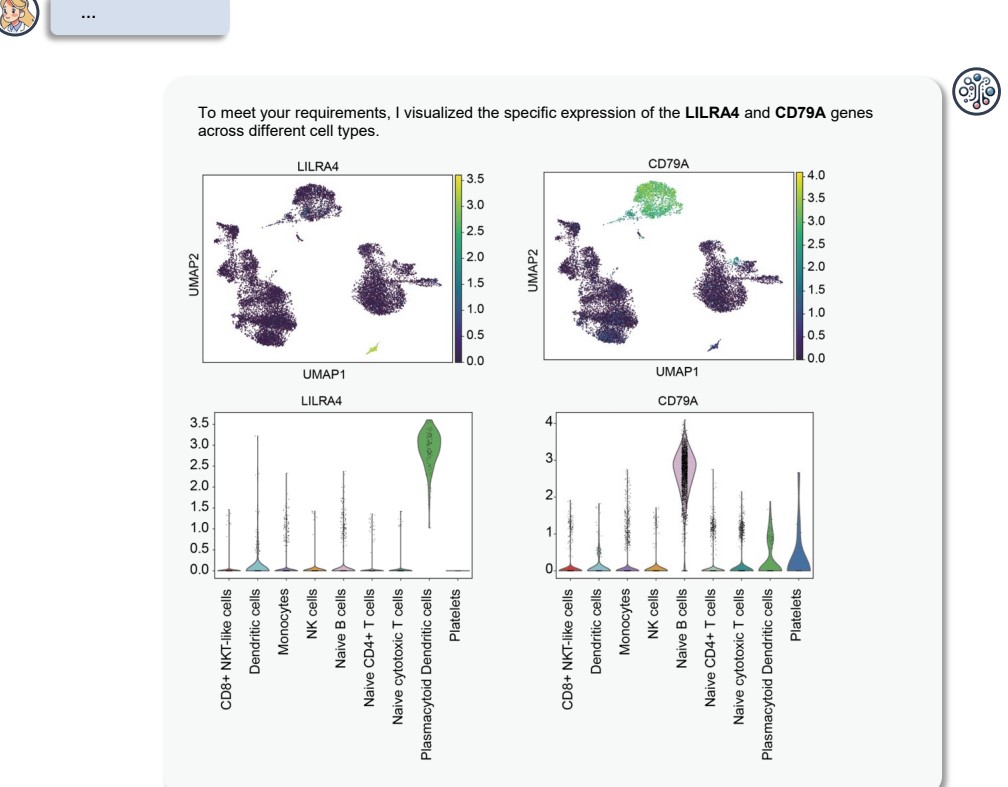

Figure 13: **Results on cell type annotation through dialogue**. After cell type annotation, CellAgent provided a visualization of LILR4A gene (PDCs marker) and CD79A gene (B cells marker) expression across clusters using UMAP plots. The violin plot further illustrates the gene expression distribution within these cell clusters.

## E.9 RESULTS ON TRAJECTORY INFERENCE THROUGH DIALOGUE

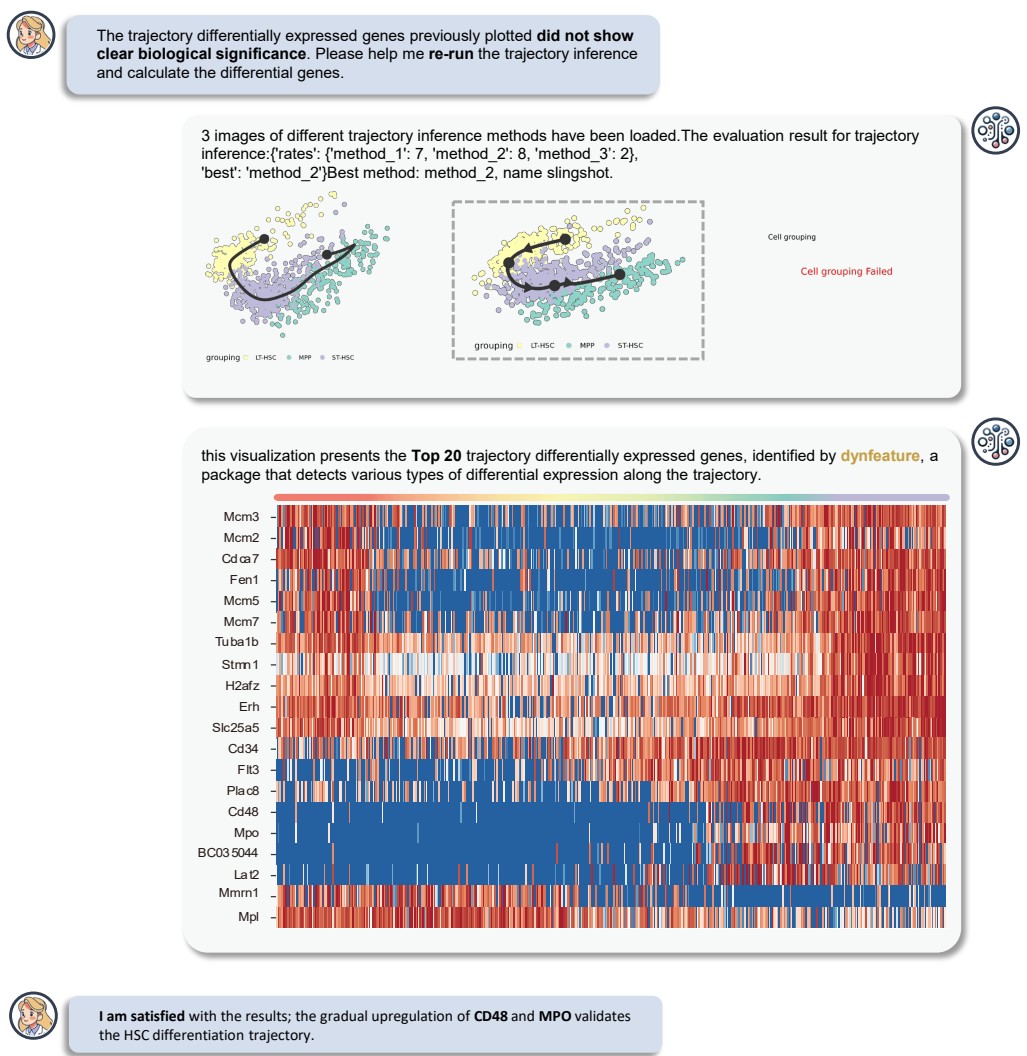

Figure 14: **Results on trajectory inference through dialogue**. In response to user dissatisfaction with the initial trajectory differential expression results, CellAgent automatically evaluated three distinct trajectory inference methods and selected the highest-scoring one (Slingshot). It then visualized the top 20 differentially expressed genes along the refined trajectory.

## E.10 THE PERFORMANCE OF DIFFERENT METHODS ON BATCH CORRECTION TASK

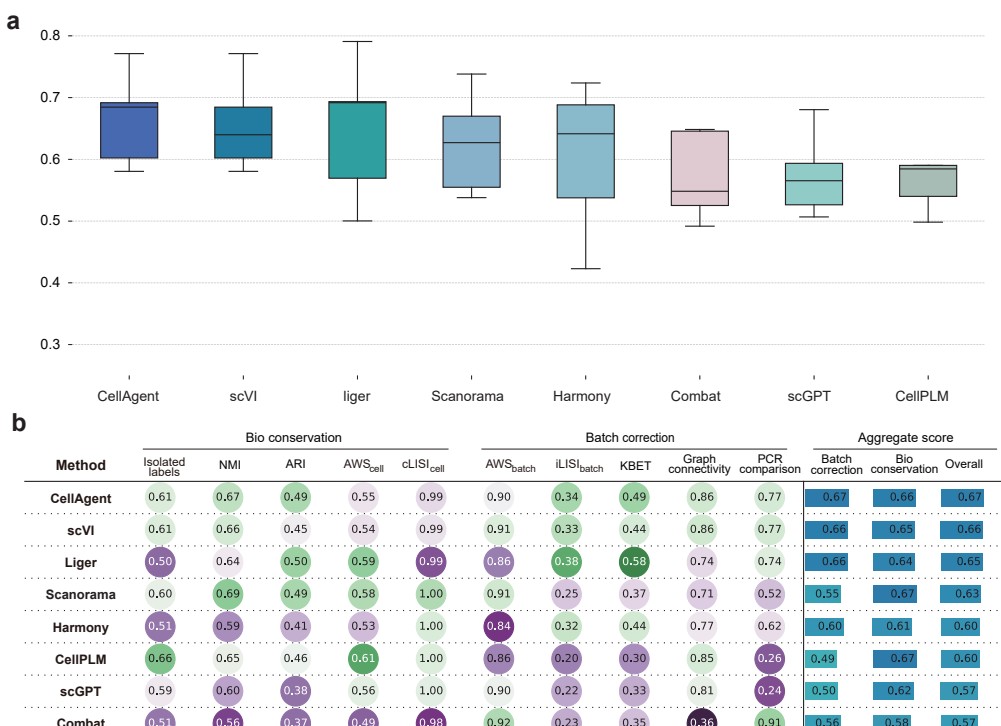

Figure 15: **The performance of different methods on batch correction task**. **a,** Violin plot shows the distribution of overall score between CellAgent and other batch correction methods across all datasets. **b,** This figure presents the benchmarking results of CellAgent alongside Liger (Liu et al., 2020), scVI (Lopez et al., 2018), Scanorama (Hie et al., 2019), Harmony (Korsunsky et al., 2019), CellPLM (Wen et al., 2024), scGPT (Cui et al., 2024), and Combat (Johnson et al., 2007) across five datasets, evaluating their average performance based on multiple metrics related to biological conservation and batch effect removal. The biological conservation assessment includes Isolated Labels (Korsunsky et al., 2019), NMI (Pedregosa et al., 2011), ARI (Hubert & Arabie, 1985), $AWS_{cell}$ (Büttner et al., 2019), and $cLISI_{cell}$ (Korsunsky et al., 2019), which measure the extent to which the biological structure is preserved after integration. The batch correction evaluation comprises Graph Connectivity (Luecken et al., 2022), PCR Comparison (Luecken et al., 2022), KBET (Luecken et al., 2022), $iLISI_{batch}$ (Korsunsky et al., 2019), and $AWS_{batch}$ (Büttner et al., 2019), reflecting the effectiveness of removing batch effects. The results are visualized using heatmaps. The aggregate scores on the rightmost panel summarize the overall performance of each method, with batch correction and biological conservation scores weighted to compute the final overall score.

### E.11 THE UMAP VISUALIZATION OF CELLAGENT ON FIVE DIFFERENT DATASETS, COLORED BY BATCH LABELS

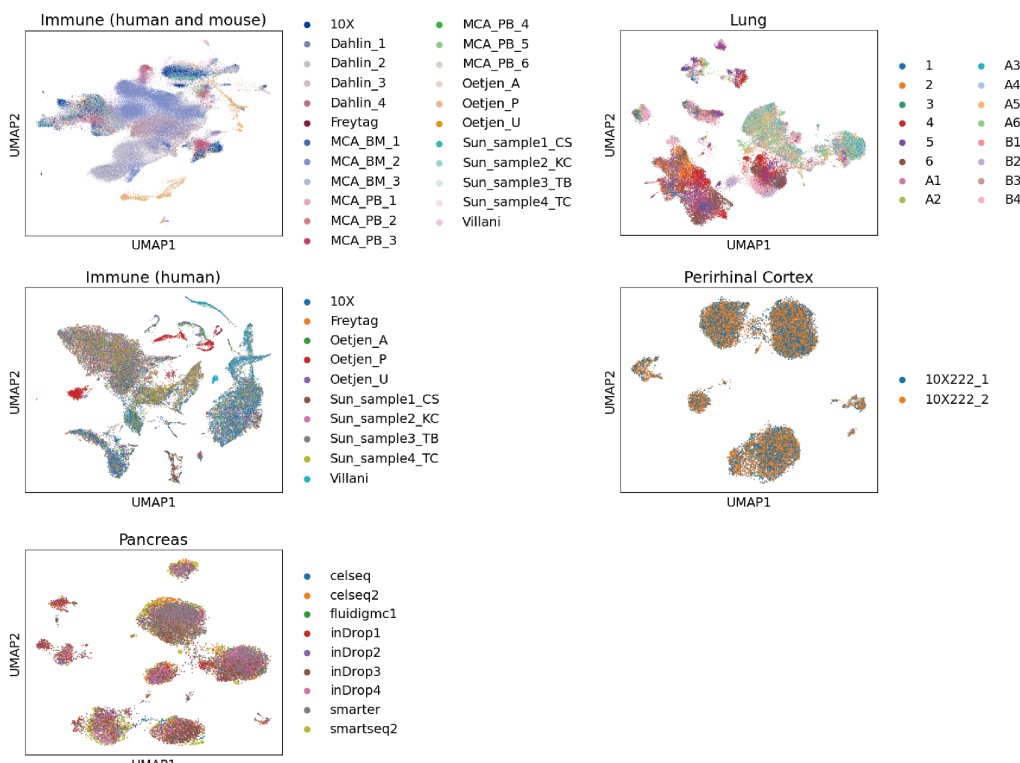

Figure 16: **The UMAP visualization of CellAgent on five different datasets, colored by batch labels**. Each point represents a cell, and the colors correspond to different batch labels, indicating the source of the samples in each dataset. The visualization demonstrates the clustering of cells across different experimental conditions.

### E.12 The UMAP visualization of CellAgent on five different datasets, colored by cell type labels

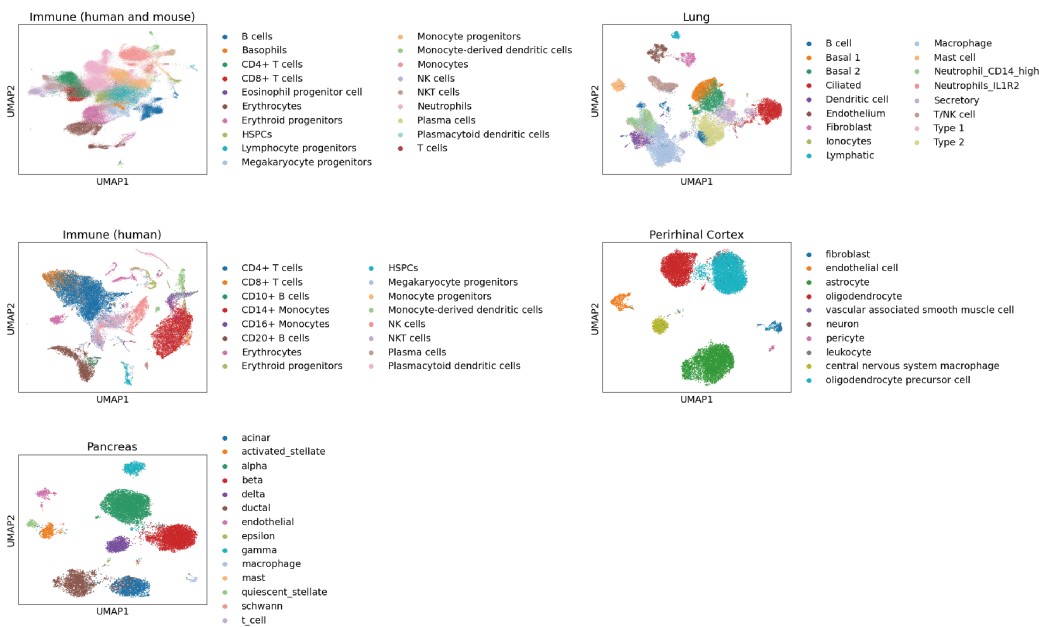

Figure 17: **The UMAP visualization of CellAgent on five different datasets, colored by cell type labels**. Each point represents a cell, and the colors correspond to different cell types identified within each dataset. The visualization demonstrates the clustering of cell across diverse biological contexts.

## E.13 THE PERFORMANCE OF DIFFERENT METHODS FOR CELL TYPE ANNOTATION ACROSS SIX DATASETS

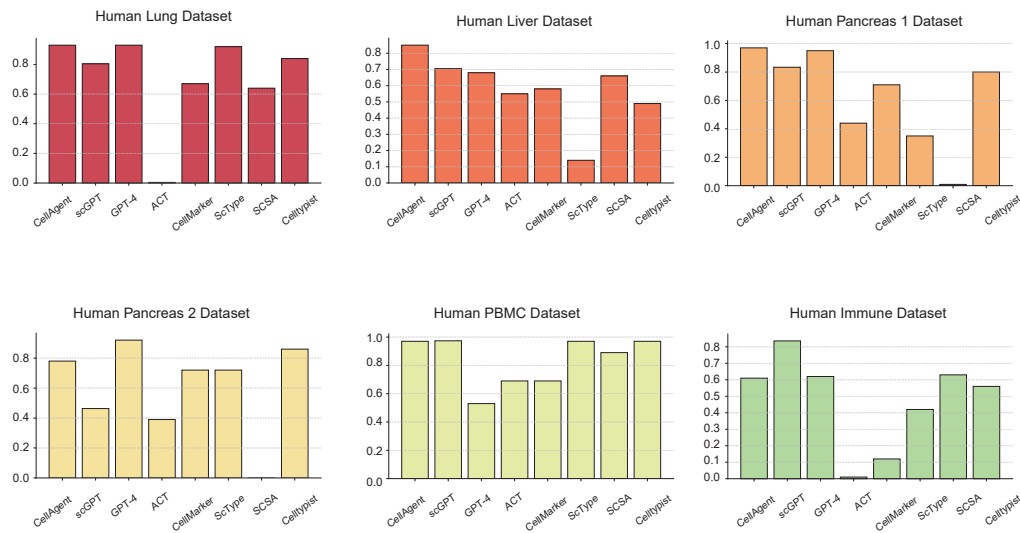

Figure 18: **The performance of different methods for cell type annotation across six datasets**. The bar chart shows the average accuracy of CellAgent compared to other different cell type annotation methods across six different datasets.

### E.14    UMAP VISUALIZATION OF CELLAGENT-INFERRED TRAJECTORIES COLORED BY PSEUDOTIME

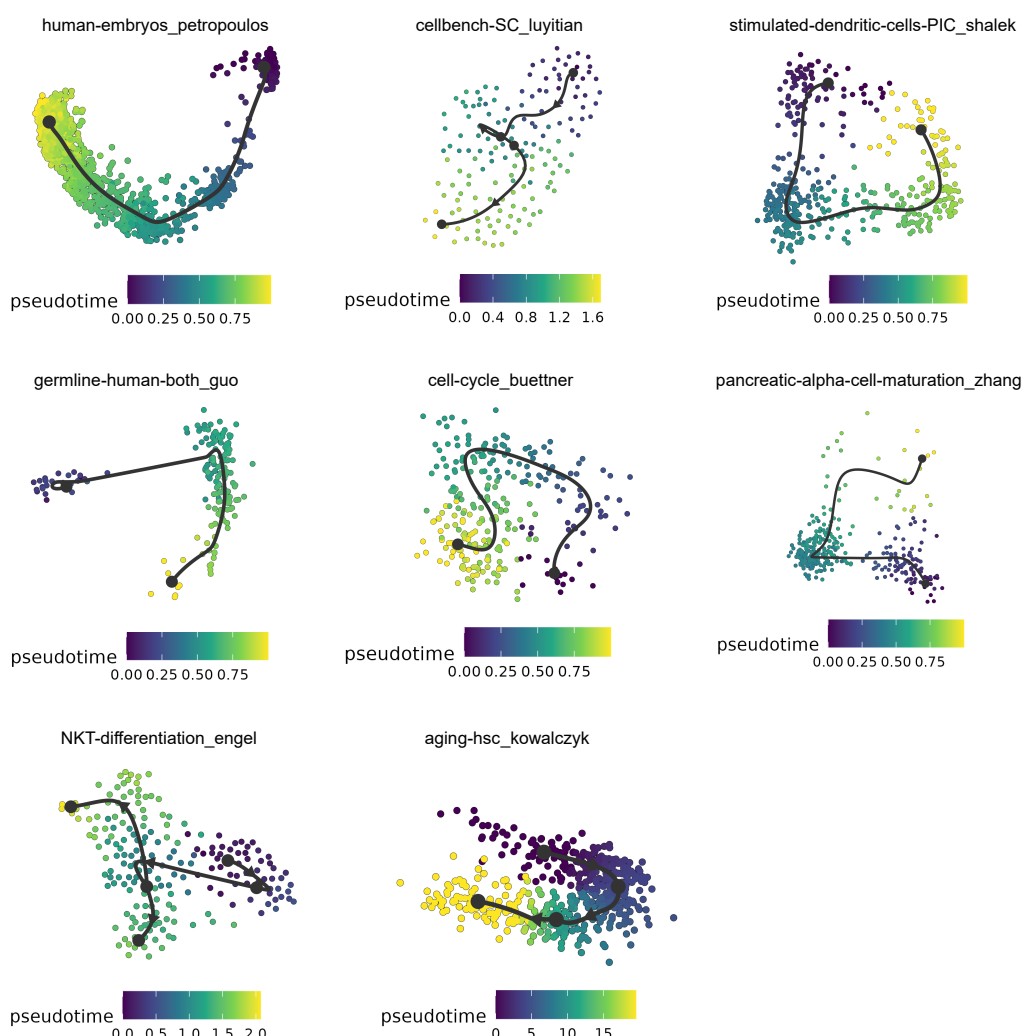

Figure 19: **UMAP visualization of CellAgent-inferred trajectories, colored by pseudotime**. Each subplot represents a distinct dataset, where dots correspond to individual cells, and colors indicate pseudotime progression, with darker colors denoting earlier states and lighter colors indicating later stages. The inferred trajectory paths capture the developmental or differentiation progression within each dataset.

### E.15 UMAP VISUALIZATION OF CELLAGENT-INFERRED TRAJECTORIES COLORED BY CELL TYPE

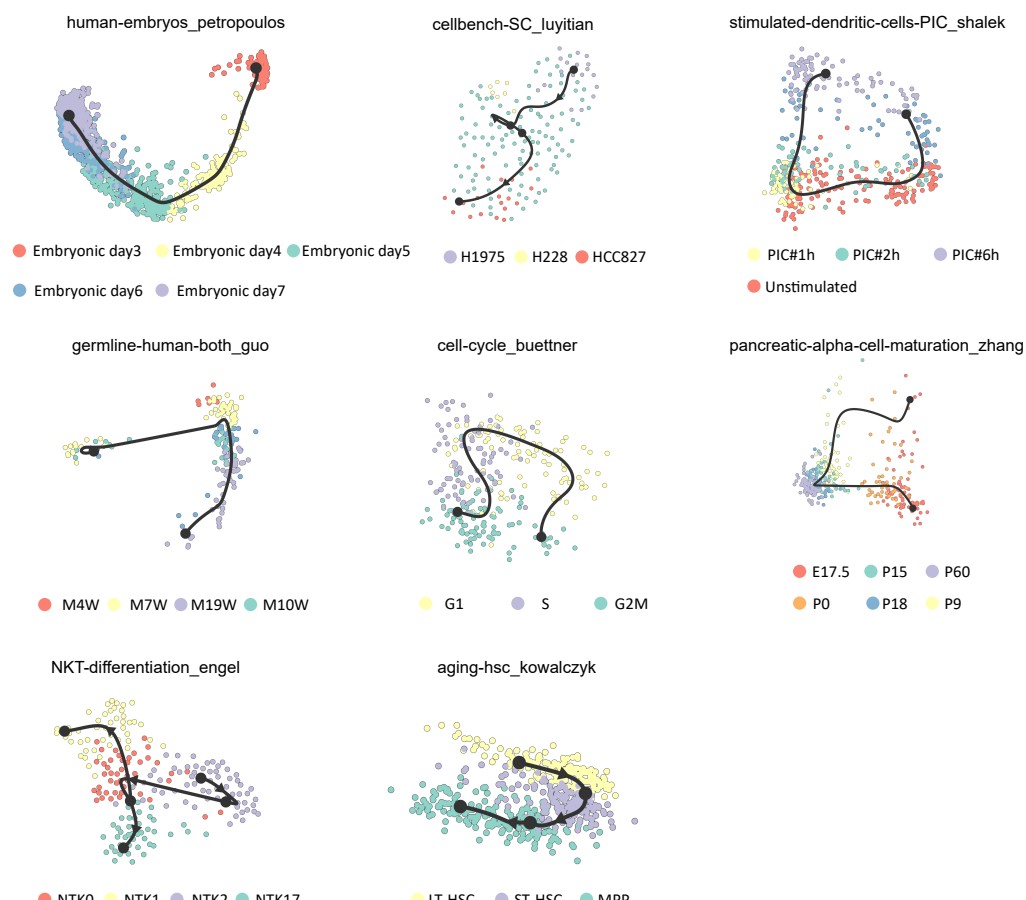

Figure 20: **UMAP visualization of CellAgent-inferred trajectories, colored by cell type.** Each subplot represents a distinct dataset, where dots correspond to individual cells, and colors indicate cell type. The trajectory covered various biological processes and the inferred trajectories capture the developmental or differentiation pathways within each dataset.

## E.16    VISUALIZATION OF DLPFC SLICES DATASETS

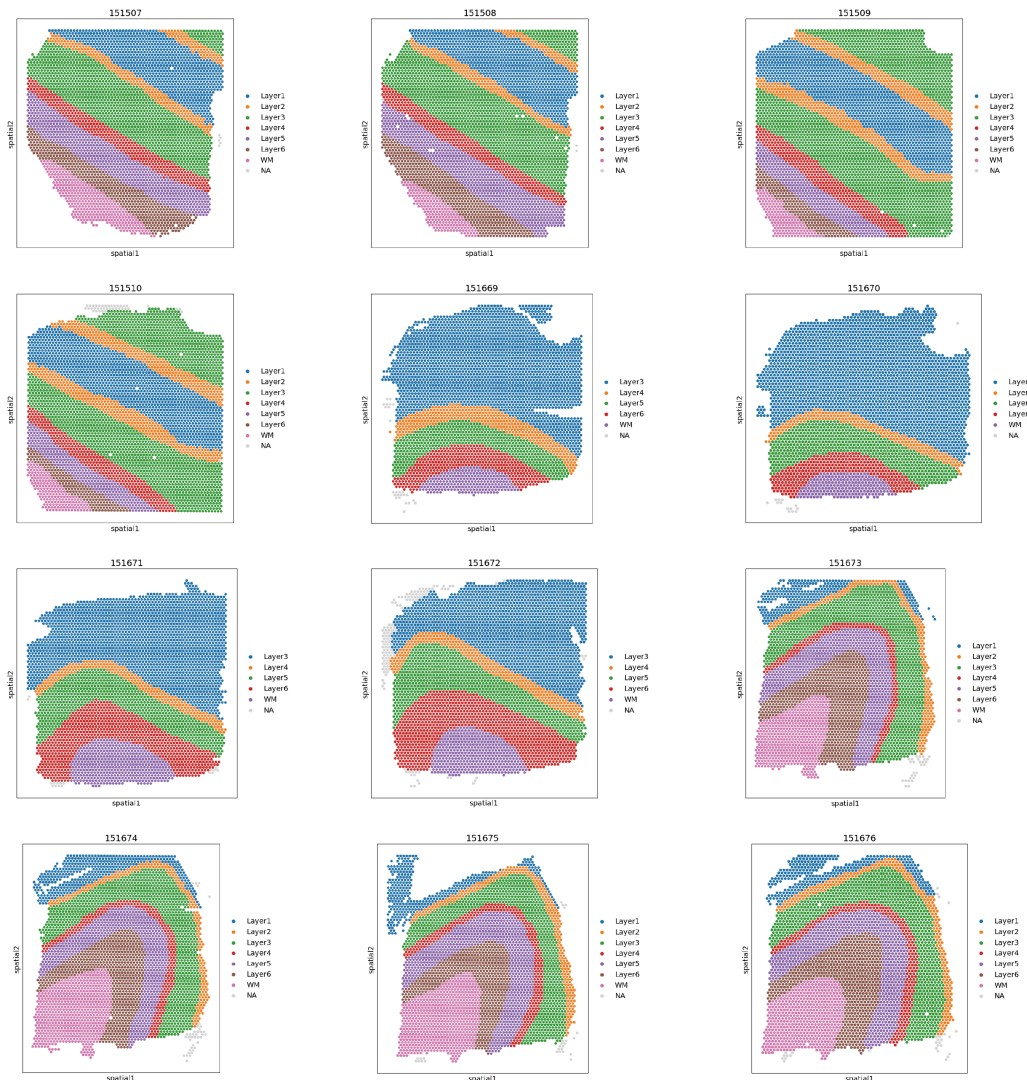

Figure 21: **Visualization of DLPFC slices datasets.** The DLPFC layers were annotated by Maynard et al. (Maynard et al., 2021), where the ground truth spots in slices were mapped to their spatial positions and classified into six cortical layers (L1–L6) and white matter (WM), NA indicates spots that were not annotated.

## E.17 PREDICTED SPATIAL DOMAINS OF CELLAGENT ON DLPFC DATASETS

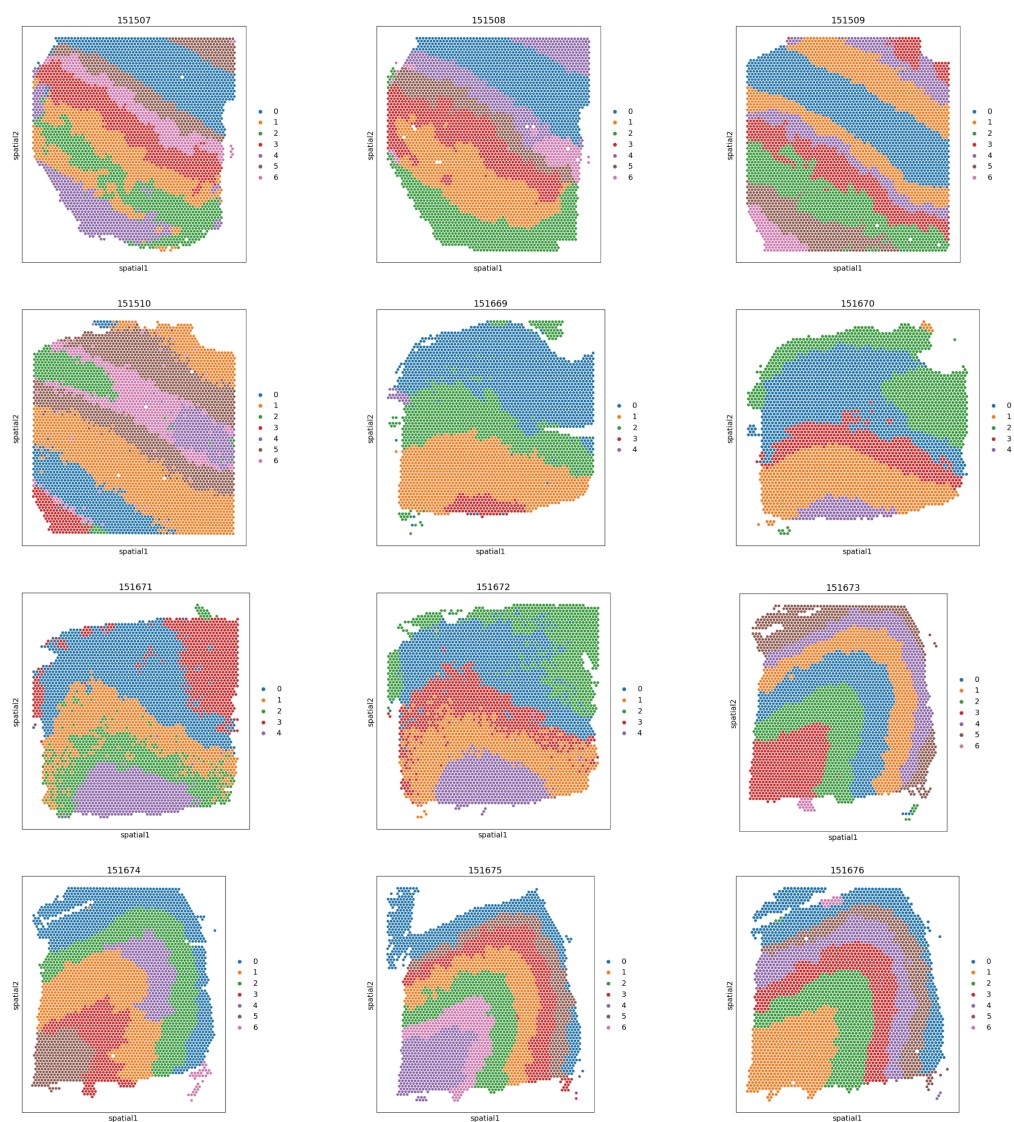

Figure 22: **Predicted spatial domains of CellAgent on DLPFC datasets.** The figures present the predicted spatial domains of CellAgent across DLPFC slices datasets, with spots colored according to the inferred spatial domains.

# F   SUPPLEMENTARY TABLES

## F.1   THE OVERVIEW OF THE SC-OMNI TOOLKIT

| Task Name | Task Category | Tools Involved |
|---|---|---|
| Removal of Low-Quality Cells and Genes | Preliminary Analysis | scanpy |
| Expression Normalization | Preliminary Analysis | scanpy |
| Identification of Highly Variable Genes | Preliminary Analysis | scanpy |
| Dimensionality Reduction | Essential Analysis | PCA_scanpy, UMAP_scanpy, t-SNE_scanpy, etc. |
| Neighbor Graph Computation | Essential Analysis | scanpy, squidpy |
| Clustering | Essential Analysis | louvain_scanpy, leiden_scanpy, KMeans_scanpy, etc. |
| Cell Type Annotation | Essential Analysis | Celltypist, ScType, CellMarker2.0, etc. |
| Differentially Expressed Gene Identification | Advanced Analysis | wilcoxon_scanpy, logreg_scanpy, t-test_scanpy, etc. |
| Batch Correction | Advanced Analysis | Harmony, scVI, Scanorama, etc. |
| Trajectory Inference | Advanced Analysis | PAGA_Dyno, Slingshot_Dyno, Scorpius_Dyno, etc. |
| Imputation for Spatial Transcriptomics | Advanced Analysis | Tangram, SpaGE, novoSpaRc, etc. |
| Spatial Trajectory Inference | Advanced Analysis | stLearn |
| Identification of Spatially Variable Genes | Essential Analysis | squidpy |
| Compute Ripley's Statistics | Essential Analysis | squidpy |
| Spatial Domain Identification | Advanced Analysis | DeepST, SEDR, SpaGCN, etc. |
| Co-occurrence Probability | Essential Analysis | squidpy |
| Neighborhood Enrichment | Essential Analysis | squidpy |

Table 2: **The overview of the sc-Omni toolkit.** Each task is categorized into preliminary, essential, or advanced analysis, with representative tools or algorithms.

## F.2 OVERVIEW OF THE EIGHT MAJOR TASKS IN SCRNA-SEQ AND SPATIAL TRANSCRIPTOMICS ANALYSES

| Task | Description | Purpose |
|---|---|---|
| Task 1 | Batch correction | Remove batch effects while conserving biological variation. |
| Task 2 | Preprocessing and clustering | Remove low-quality cells and genes, perform normalization, identify highly variable genes, apply dimensionality reduction, and finally cluster cells with similar states. |
| Task 3 | Cell type annotation | Identify the cell types of each cluster. |
| Task 4 | Trajectory inference | Reconstruct the dynamic developmental trajectory of cells by ordering them along the pseudotime. |
| Task 5 | Spatial neighborhood analysis | Reveal spatial relationships and interactions between cell clusters in the tissue or organ. |
| Task 6 | Spatially variable gene identification | Identify genes with distinct spatial expression patterns in tissues or organs. |
| Task 7 | Spatial transcriptomics imputation | Predict gene expression levels in spatial transcriptomics data by leveraging reference single-cell datasets. |
| Task 8 | Spatial domain identification | Partition regions with similar gene expression patterns and tissue structures into distinct spatial domains through spatial clustering. |

Table 3: **Overview of the eight major tasks in scRNA-seq and spatial transcriptomics analyses.** These tasks include batch correction, preprocessing and clustering, cell type annotation, trajectory inference, spatial neighborhood analysis, spatially variable gene identification, spatial transcriptomics imputation, and spatial domain identification. The table also outlines the primary objectives of each task.

## F.3 OVERVIEW OF BATCH CORRECTION DATASETS

| Batch Correction Task | Cell Number | Batches | Tested Features |
|---|---|---|---|
| Pancreas | 16382 | 9 batches | Widely used test data, protocols |
| Lung | 42000 | 4 batches | Human variation, protocols, spatial locations, high resolution scoping, tissue types |
| Perirhinal Cortex | 17353 | 2 batches | Brain region specificity, diverse gene expression profiles, precise cell-type annotations |
| Immune (human) | 35206 | 10 batches | Tissues, laboratories, cell types |
| Immune (human and mouse) | 97552 | 23 samples | Tissues, laboratories, similar cell types, species |

Table 4: **Overview of batch correction datasets.** The table summarizes the key characteristics of the five datasets used for batch correction evaluation, including the number of cells and batches, as well as the tested features such as tissue types, experimental protocols, and laboratory sources.

## F.4 OVERVIEW OF CELL TYPE ANNOTATION DATASETS

| Cell Type Annotation Task | Cell Number | Gene Number | Cell Type Count |
|---|---|---|---|
| PBMC | 15476 | 33694 | 9 |
| Pancreas 1 | 2126 | 34290 | 10 |
| Pancreas 2 | 8569 | 34290 | 14 |
| Liver | 6439 | 47951 | 27 |
| Immune | 33506 | 12303 | 16 |
| Lung | 32472 | 15148 | 17 |

Table 5: **Overview of cell type annotation datasets.** The table summarizes the key characteristics of the datasets used for cell type annotation evaluation, including the number of cells, the number of genes, and the number of cell types in each dataset.

## F.5 OVERVIEW OF TRAJECTORY INFERENCE DATASETS

| Trajectory Inference Task | Cell Number | Gene Number | Trajectory Type |
|---|---|---|---|
| Aging-hsc Kowalczyk | 493 | 2227 | Linear |
| Human-embryos Petropoulos | 1289 | 8772 | Linear |
| Cell-cycle-hESCs Leng | 1544 | 1770 | Cycle |
| Germline-human-both Guo | 2202 | 8148 | Bifurcation |
| Cell-cycle Buettner | 264 | 6812 | Cycle |
| NKT-differentiation Miao | 3999 | 7799 | Multifurcation |
| Simulated-dendritic-cells-PIC Shackel | 4332 | 4158 | Linear |
| Pancreatic-alpha-cell-maturation Zhang | 322 | 6138 | Linear |

Table 6: **Overview of trajectory inference datasets.** The table summarizes the key characteristics of the datasets used for trajectory inference evaluation, including the number of cells, gene count, and the trajectory type (e.g., linear, bifurcation, cycle, or multifurcation) for each dataset.

## F.6 OVERVIEW OF SPATIAL DOMAIN IDENTIFICATION DATASETS

| Spatial Domain Identification Task | Spot Number | Gene Number | Domain Number |
|---|---|---|---|
| Slice 151507 | 4226 | 33538 | 7 |
| Slice 151508 | 4093 | 33538 | 7 |
| Slice 151509 | 4784 | 33538 | 7 |
| Slice 151510 | 4369 | 33538 | 7 |
| Slice 151511 | 3861 | 33538 | 5 |
| Slice 151512 | 3606 | 33538 | 5 |
| Slice 151671 | 4221 | 33538 | 7 |
| Slice 151672 | 3849 | 33538 | 7 |
| Slice 151673 | 3692 | 33538 | 7 |
| Slice 151674 | 3590 | 33538 | 7 |
| Slice 151675 | 3632 | 33538 | 7 |

Table 7: **Overview of spatial domain identification datasets.** The table summarizes the key characteristics of the DLPFC datasets used for spatial domain identification evaluation, including the number of spots, gene count, and the number of spatial domains for each slice.

## F.7 OVERVIEW OF SPATIAL IMPUTATION DATASETS

| Spatial Imputation Task | Spot Number | Gene Number | Technology |
|---|---|---|---|
| Embryo | 15476 | 33694 | seqFISH (image-based) |
| Cortex | 5240 | 10000 | seqFISH (image-based) |
| Prefrontal cortex | 8569 | 3429 | STARmap (image-based) |
| Embryo | 6439 | 47951 | seqFISH (image-based) |
| Breast cancer | 3326 | 16384 | Visium (seq-based) |
| Kidney | 3570 | 15148 | Visium (seq-based) |
| Hippocampus | 2044 | 3105 | Slide-seqV2 (seq-based) |

Table 8: **Overview of spatial imputation datasets.** The table summarizes the key characteristics of the datasets used for spatial imputation evaluation, including the number of spots, gene count, and the technologies used (e.g., image-based or seq-based).

## F.8 OVERVIEW OF DATASETS USED IN EXPERIMENTS

| Dataset Number | Dataset Description | Dataset Source |
| --- | --- | --- |
| Dataset 1 | Human blood atlas | Xu et al. (Xu et al., 2023) |
| Dataset 2 | Human bone marrow atlas | Xu et al. (Xu et al., 2023) |
| Dataset 3 | Human heart atlas | Xu et al. (Xu et al., 2023) |
| Dataset 4 | Human intestine atlas | Xu et al. (Xu et al., 2023) |
| Dataset 5 | Human kidney atlas | Xu et al. (Xu et al., 2023) |
| Dataset 6 | Human liver atlas | Xu et al. (Xu et al., 2023) |
| Dataset 7 | Human lung atlas | Xu et al. (Xu et al., 2023) |
| Dataset 8 | Human pancreas atlas | Xu et al. (Xu et al., 2023) |
| Dataset 9 | Human skeletal muscle atlas | Xu et al. (Xu et al., 2023) |
| Dataset 10 | Human spleen atlas | Xu et al. (Xu et al., 2023) |
| Dataset 11 | Human pancreas | Luecken et al. (Luecken et al., 2022) |
| Dataset 12 | Human lung atlas | Luecken et al. (Luecken et al., 2022) |
| Dataset 13 | Immune cells (human) | Luecken et al. (Luecken et al., 2022) |
| Dataset 14 | Immune cells (human&mouse) | Luecken et al. (Luecken et al., 2022) |
| Dataset 15 | Human perirhinal cortex | Siletti et al. (Siletti et al., 2023) |
| Dataset 16 | PBMC | Zheng et al. (Zheng et al., 2017) |
| Dataset 17 | Human pancreas 1 | Baron et al (Baron et al., 2016) |
| Dataset 18 | Human pancreas 2 | Muraro et al. (Muraro et al., 2016) |
| Dataset 19 | Human pancreas 3 | Lawlor et (Lawlor et al., 2017) |
| Dataset 20 | Human pancreas 4 | Segerstolpe et al. (Segerstolpe et al., 2016) |
| Dataset 21 | Human pancreas 5 | xin et al (Xin et al., 2016) |
| Dataset 22 | Human liver | Guilliams et al. (Guilliams & Scott, 2022) |
| Dataset 23 | Mouse retina | Macosko et al. (Macosko et al., 2015) |
| Dataset 24 | Hypothalamic preoptic region | Moffitt et al. (Moffitt et al., 2018) |
| Dataset 25 | Mouse brain section (coronal) | Zhou et al. (Zhou et al., 2020) |
| Dataset 26 | Mouse cortex | Tasic et al. (Tasic et al., 2018) |
| Dataset 27 | Myeloid progenitors | Paul et al. (Paul et al., 2015) |
| Dataset 28 | Human hindlimb muscle | McKellar et al (McKellar et al., 2020). |
| Dataset 29 | Human prostate | McCray et al. (McCray et al., 2021) |
| Dataset 30 | Human osteosarcoma | Zhou et al. (Zhou et al., 2020) |
| Dataset 31 | Aging-hsc Kowalczyk | Saelens et al. (Saelens et al., 2019) |
| Dataset 32 | Human-embryos Petropoulos | Saelens et al. (Saelens et al., 2019) |
| Dataset 33 | Cellbench-SC luyitian | Saelens et al. (Saelens et al., 2019) |
| Dataset 34 | Germline-human-both guo | Saelens et al. (Saelens et al., 2019) |
| Dataset 35 | Cell-cycle Buettner | Saelens et al. (Saelens et al., 2019) |
| Dataset 36 | NKT-differentiation engel | Saelens et al. (Saelens et al., 2019) |
| Dataset 37 | Stimulated-dendritic-cells-PIC shalek | Saelens et al. (Saelens et al., 2019) |
| Dataset 38 | Pancreatic-alpha-cell-maturation zhang | Saelens et al. (Saelens et al., 2019) |
| Dataset 39 | DLPFC, slice 151507 | Maynard et al. (Maynard et al., 2021) |
| Dataset 40 | DLPFC, slice 151508 | Maynard et al. (Maynard et al., 2021) |
| Dataset 41 | DLPFC, slice 151509 | Maynard et al. (Maynard et al., 2021) |
| Dataset 42 | DLPFC, slice 151510 | Maynard et al. (Maynard et al., 2021) |
| Dataset 43 | DLPFC, slice 151669 | Maynard et al. (Maynard et al., 2021) |

| Dataset 44 | DLPFC, slice 151670 | Maynard et al. (Maynard et al., 2021) |
|---|---|---|
| Dataset 45 | DLPFC, slice 151671 | Maynard et al. (Maynard et al., 2021) |
| Dataset 46 | DLPFC, slice 151672 | Maynard et al. (Maynard et al., 2021) |
| Dataset 47 | DLPFC, slice 151673 | Maynard et al. (Maynard et al., 2021) |
| Dataset 48 | DLPFC, slice 151674 | Maynard et al. (Maynard et al., 2021) |
| Dataset 49 | DLPFC, slice 151675 | Maynard et al. (Maynard et al., 2021) |
| Dataset 50 | DLPFC, slice 151676 | Maynard et al. (Maynard et al., 2021) |
| Dataset 51 | Embryonic (scRNA-seq) | Li et al. (Li et al., 2022) |
| Dataset 52 | Cortex (scRNA-seq) | Li et al. (Li et al., 2022) |
| Dataset 53 | Embryo (scRNA-seq) | Li et al. (Li et al., 2022) |
| Dataset 54 | Prefrontal cortex (scRNA-seq) | Li et al. (Li et al., 2022) |
| Dataset 55 | Breast cancer (scRNA-seq) | Li et al. (Li et al., 2022) |
| Dataset 56 | Kidney (scRNA-seq) | Li et al. (Li et al., 2022) |
| Dataset 57 | Hippocampus (scRNA-seq) | Li et al. (Li et al., 2022) |
| Dataset 58 | Embryonic (seqFISH) | Li et al. (Li et al., 2022) |
| Dataset 59 | Cortex (seqFISH+) | Li et al. (Li et al., 2022) |
| Dataset 60 | Embryo (STARmap) | Li et al. (Li et al., 2022) |
| Dataset 61 | Prefrontal cortex (FISH) | Li et al. (Li et al., 2022) |
| Dataset 62 | Breast cancer (Visium) | Li et al. (Li et al., 2022) |
| Dataset 63 | Kidney (Visium) | Li et al. (Li et al., 2022) |
| Dataset 64 | Hippocampus (Slide-seqV2) | Li et al. (Li et al., 2022) |

Table 9: **Summary of the datasets utilized in scRNA-seq and spatial transcriptomics analyses.**

