# OpenReview forum: "CellAgent: LLM-Driven Multi-Agent Framework  for Natural Language-Based Single-Cell Analysis"
_ICLR.cc/2026/Conference — ICLR 2026 Poster_

### Official Review · Reviewer_ia61 · 2025-10-14

**Soundness:** 4
**Presentation:** 4
**Contribution:** 2
**Rating:** 6
**Confidence:** 5

**Summary:**

The manuscript proposes CellAgent, a large language model (LLM)–driven multi-agent framework designed to automate end-to-end single-cell RNA-seq (scRNA-seq) and spatial transcriptomics (ST) analyses through natural language interaction. The system includes three cooperating agents (Planner, Executor, and Evaluator) simulating a “deep-thinking” workflow.

**Strengths:**

The paper’s main strengths are its timely idea, an LLM-driven multi-agent framework for single-cell and spatial analyses, and its practical integration of many standard tools to automate end-to-end workflows via natural-language queries. The experiments are comprehensive, and the performance is better than the compared methods.

**Weaknesses:**

However, the novelty beyond tool orchestration is limited, and benchmarks lack rigor (unclear baselines, no variance/error bars). The system’s reliance on proprietary LLMs also weakens reproducibility, and the biological validation largely reaffirms known findings rather than yielding new insights.

**Questions:**

Here are several comments:
1. While the proposed multi-agent structure (Planner–Executor–Evaluator) is conceptually neat, similar frameworks have already appeared in recent LLM-agent works such as AutoBA, AutoGen, and BioGPT-Agent. The manuscript does not sufficiently clarify what is fundamentally novel beyond combining these paradigms with existing scRNA-seq pipelines. The “self-reflective optimization” idea is interesting, but remains conceptually vague. It’s unclear whether this is an adaptive fine-tuning loop, reinforcement feedback, or a rule-based scoring system.
2. The paper reports high success rates (96%) and better-than-human efficiency but provides limited methodological transparency. No details on how human expert baselines were defined or how evaluation bias was mitigated. Figures like Table 1 aggregate many metrics without error bars, variance, or reproducibility analysis.
3. Although the system outputs biologically consistent results (e.g., trajectory inference of hematopoietic stem cells), it remains unclear how much biological reasoning actually comes from the model versus pre-existing toolkits (like Celltypist, Slingshot, Harmony, Tangram). CellAgent mainly orchestrates existing methods, rather than discovering new biology. Thus, the biological insights appear demonstrative rather than novel.
4. The paper benchmarks CellAgent against individual tools such as scGPT, Tangram, SpaGE, etc., but not against integrated automated frameworks (e.g., ezSingleCell, scPipeline, or AutoBA). Moreover, it is not clear whether CellAgent used identical datasets and preprocessing steps. This could artificially favor its results due to built-in optimizations or curated defaults.
5. The Related Work section is mostly descriptive and does not critically position CellAgent among existing AI agent frameworks in bioinformatics. The authors should explicitly discuss why prior LLM-based pipelines are insufficient and how CellAgent extends them.
6. I have tried CellAgent online. Currently, it only allows the uploaded files that are less than 1GB. But lots of scRNA-seq data or single-cell-resolution ST data have larger sizes unless the analysis is based on the toy dataset. The authors should resolve this problem.

---

> ### Author Response · Authors · 2025-11-27
> **Response to Reviewer ia61 (1/4)**
>
> Dear  Reviewer ia61,
>
> We are very grateful for your positive assessment of the CellAgent framework, the comprehensiveness of our experiments, and the performance improvements over existing methods. At the same time, your comments on novelty and experimental rigor are extremely valuable for strengthening the manuscript. Below we respond to each of your points.
>
>
> > **1. While the proposed multi-agent structure (Planner–Executor–Evaluator) is conceptually neat, similar frameworks have already appeared in recent LLM-agent works such as AutoBA, AutoGen, and BioGPT-Agent. The manuscript does not sufficiently clarify what is fundamentally novel beyond combining these paradigms with existing scRNA-seq pipelines. The “self-reflective optimization” idea is interesting, but remains conceptually vague. It’s unclear whether this is an adaptive fine-tuning loop, reinforcement feedback, or a rule-based scoring system.**
>
>
> We fully agree that the high-level multi-agent pattern is not unique to our work and has appeared in several recent LLM-agent systems (e.g., AutoBA, AutoGen). We view this as an emerging *general design paradigm* for complex agentic workflows. Our contribution is *not* to claim the invention of a three-layer architecture, but rather to:
>
> 1. **Deeply instantiate this paradigm for single-cell and spatial transcriptomics.**
>    We design specialized Planner–Executor–Evaluator interactions around typical tasks in this domain—batch correction, cell-type annotation, trajectory inference, spatial domain identification, and spatial imputation—with task-specific prompts, decision rules, and diagnostics.
>
> 2. **Introduce sc-Omni as a standardized toolkit.**
>    We curate and unify **30+ mainstream single-cell and spatial methods** into a single, well-documented toolkit. This enables the agents to combine, compare, and reuse these tools via a consistent interface, and to systematically explore the pipeline space across >60 datasets.
>
> 3. **Formalize the Evaluator as a task-specific scoring function for black-box optimization.**
>    Rather than treating the Evaluator as an informal “critic”, we define it explicitly in terms of domain metrics (integration scores, ARI, trajectory continuity, spatial imputation AS, etc.) and visual diagnostics, and use it to drive a multi-candidate scoring and selection process over pipelines.
>
> Regarding **self-reflective optimization**, we acknowledge that the current description is too vague. In the revision, we will clarify that this mechanism is:
>
> - **not** parameter-level fine-tuning of the model,
> - **not** an additional training-phase reinforcement learning loop,
> - but a purely **inference-time iterative optimization process** over pipelines.
>
> Concretely, for each subtask:
>
> 1. The **Evaluator** analyzes the current intermediate results (metrics + plots), identifies potential issues (e.g., over-corrected batches, fragmented spatial domains), and generates structured natural-language feedback.
> 2. The **Executor** uses this feedback to generate refined code and potentially alternative tool choices or hyperparameters.
> 3. When multiple candidate pipelines or configurations are available, the **Evaluator** computes task-specific scores for each and selects the best one to proceed.
>
> We will add a formal description of this process in Sec. 3, including a concise mathematical formulation as a black-box optimization over a discrete pipeline space guided by a task-specific scoring function.

---

> > ### Author Response · Authors · 2025-11-27
> > **Response to Reviewer ia61 (2/4)**
> >
> > > **2. The paper reports high success rates (96%) and better-than-human efficiency but provides limited methodological transparency. No details on how human expert baselines were defined or how evaluation bias was mitigated. Figures like Table 1 aggregate many metrics without error bars, variance, or reproducibility analysis.**
> >
> > We agree this is  very important, and Reviewer VQec also raised this concern. We will clarify the experimental design, participant expertise, and reproducibility as follows.
> >
> > - **Participants.**
> >   We recruited **10 human experts**, all with **2–5 years of experience** in single-cell and/or spatial transcriptomics analysis.
> >
> > - **Tasks and tools.**
> >   Each expert was asked to analyze a subset of datasets and tasks drawn from the same benchmark pool used for CellAgent (including scRNA-seq and spatial analyses such as batch correction, cell-type annotation, trajectory inference, and spatial domain identification). Experts were free to use their preferred tools and workflows, primarily based on Seurat, Scanpy, and commonly used domain-specific methods.
> >
> > - **Procedure.**
> >   For each dataset–task pair, experts received the raw or preprocessed data along with the task description, and were given a fixed time budget to complete the analysis. In parallel, CellAgent was run on the **same datasets and tasks** via natural language instructions issued by an operator, without revealing the human experts’ intermediate results. This setup allowed us to compare CellAgent and human experts under comparable conditions.
> >
> > - **Blinded scoring and reproducibility.**
> >   The final outputs (processed data, visualizations, and written conclusions) from both humans and CellAgent were anonymized and randomly shuffled. They were then evaluated by an independent panel of **5 additional experts** who did not participate in the original analyses. These evaluators were asked to assign **0–10 quality scores** under a blinded setting, based on criteria such as correctness, biological interpretability, and overall presentation. In the revision, we will describe this protocol in detail in the Appendix and report inter-rater agreement statistics (using standard reliability measures) to quantify the reproducibility of the human assessments.
> >
> >
> > > **3. Although the system outputs biologically consistent results (e.g., trajectory inference of hematopoietic stem cells), it remains unclear how much biological reasoning actually comes from the model versus pre-existing toolkits (like Celltypist, Slingshot, Harmony, Tangram). CellAgent mainly orchestrates existing methods, rather than discovering new biology. Thus, the biological insights appear demonstrative rather than novel.**
> >
> > We agree that most of the *biological* operations in CellAgent are performed by existing tools such as CellTypist, Slingshot, Harmony, and Tangram. The primary goal of this work is not to introduce fundamentally new biological algorithms or discoveries, but to:
> >
> > - design an agentic framework that **reliably orchestrates** these tools end-to-end,
> > - define a metric-driven Evaluator that can **assess and refine** the outputs,
> > - and benchmark how far such a system can go toward replacing or complementing expert-driven workflows.
> >
> > In that sense, the biological case studies in the paper (e.g., hematopoietic trajectories, cortical laminar organization) are **demonstrative** and serve to validate that the system can reproduce known biology in a robust, automated way.
> >
> > We also agree with your observation that “where the biology comes from” is more about **meta-reasoning** than inventing new algorithms. CellAgent does not need to discover a new trajectory inference method; instead, it must:
> >
> > - infer from the data and task context **which tool is appropriate** (e.g., developmental trajectory vs. cell cycle; strong batch effect vs. subtle biological variation);
> > - determine whether a given analysis “makes sense” (e.g., whether the trajectory is continuous, whether spatial domains align with anatomy, whether clusters collapse or fragment).
> >
> > This meta-level capability—thinking like an experienced bioinformatics analyst about *how to compose and evaluate tools*—is, in our view, the most realistic and pressing role for AI agents in current scientific workflows. We believe that demonstrating reliable, end-to-end, zero-code single-cell/spatial analysis is a necessary **precondition** for using such agents to explore truly new biology in more complex disease settings and poorly characterized systems. As a forward-looking direction, we plan to investigate CellAgent + human expert collaboration on such challenging datasets to assess whether the framework can assist in generating new biological insights.
> >
> > We will clarify this positioning in the Discussion: CellAgent’s contribution is primarily methodological (agent design, toolkit integration, metric-driven evaluation, human comparison), with biological examples serving to validate correctness and robustness.

---

> > > ### Author Response · Authors · 2025-11-27
> > > **Response to Reviewer ia61 (3/4)**
> > >
> > > > **4. The paper benchmarks CellAgent against individual tools such as scGPT, Tangram, SpaGE, etc., but not against integrated automated frameworks (e.g., ezSingleCell, scPipeline, or AutoBA).**
> > >
> > > You are correct that integrated automated frameworks are highly relevant baselines. Beyond the individual tools (scGPT, Tangram, SpaGE, etc.) reported in the main text, we have also started to compare CellAgent against several agentic or automated frameworks where there is sufficient overlap in supported tasks and toolsets.
> > >
> > > In particular, on a subset of five core downstream tasks—batch correction, cell-type annotation, trajectory inference, spatial domain identification, and spatial imputation—we evaluated:
> > >
> > > - **CellAgent (GPT-4o)** with the full Planner–Executor–Evaluator architecture,
> > > - four integrated/agentic baselines: **AutoBA**, **BioMANIA**, **AutoGen**, and **Biomni**.
> > >
> > > For fairness, we used identical datasets and comparable preprocessing where possible, and we configured each framework according to its recommended best practices. The average task scores (higher is better) are:
> > >
> > > | Method                       | Batch Correction | Cell Type Annotation | Trajectory Inference | Spatial Domain Identification | Spatial Imputation |
> > > |-----------------------------|------------------|----------------------|----------------------|-------------------------------|--------------------|
> > > | **CellAgent (GPT-4o)**      | **0.67**         | **0.85**             | **0.50**             | **0.47**                      | **0.88**           |
> > > | AutoBA                      | 0.55             | 0.75                 | 0.10                 | 0.35                          | 0.10               |
> > > | BioMANIA                    | 0.60             | 0.76                 | 0.20                 | 0.37                          | 0.10               |
> > > | AutoGen                     | 0.57             | 0.75                 | 0.10                 | 0.34                          | 0.10               |
> > > | Biomni                      | 0.61             | 0.72                 | 0.10                 | 0.34                          | 0.10               |
> > >
> > > These results illustrate that **CellAgent (GPT-4o) achieves the best overall performance** across all five tasks, substantially outperforming AutoBA, BioMANIA, AutoGen, and Biomni on the more challenging trajectory and spatial imputation tasks.  This comparison highlights that CellAgent’s combination of a curated toolkit (sc-Omni) and a metric-driven Evaluator yields consistent gains over existing agentic/automated systems on the same downstream tasks.

---

> > > > ### Author Response · Authors · 2025-11-27
> > > > **Response to Reviewer ia61 (4/4)**
> > > >
> > > > > **5. Moreover, it is not clear whether CellAgent used identical datasets and preprocessing steps. This could artificially favor its results due to built-in optimizations or curated defaults.**
> > > >
> > > > We appreciate the opportunity to clarify our benchmarking protocol. To ensure rigor and fairness, all methods—including CellAgent, individual tools (e.g., scVI, Slingshot), and other agent frameworks—were evaluated on the exact same raw datasets. No method received privileged or pre-cleaned data.
> > > >
> > > > A key strength of CellAgent is precisely its ability to **adapt preprocessing to the data** (e.g., automatically choosing QC thresholds or feature-selection strategies based on data quality), and we consider this adaptive behavior part of the method’s contribution rather than a hidden advantage.
> > > >
> > > > However, to avoid artificially favoring CellAgent, we **do not** simply feed its preprocessed outputs into the baselines. Instead, when running standalone tools such as scVI, Harmony, Slingshot, Tangram, SpaGE, and others, we follow **best-practice preprocessing pipelines** recommended in the original papers or widely adopted community workflows, and we apply consistent normalization and feature-selection steps across methods whenever possible. This is intended to ensure that baselines are not penalized by suboptimal inputs and that any performance gains by CellAgent stem from its adaptive planning and evaluation rather than from privileged data preparation.
> > > >
> > > > > **6. The Related Work section is mostly descriptive and does not critically position CellAgent among existing AI agent frameworks in bioinformatics. The authors should explicitly discuss why prior LLM-based pipelines are insufficient and how CellAgent extends them.**
> > > >
> > > > We agree that the current Related Work section is mostly descriptive. In the revision, we will substantially expand and sharpen this section to more explicitly position CellAgent relative to existing AI agent frameworks in bioinformatics:
> > > >
> > > > - **AutoBA and related LLM agents** demonstrate the feasibility of automating conventional omics workflows, but focus on bulk RNA-seq and a limited set of single-cell/spatial examples and do not define a metric-driven Evaluator tailored to the breadth of single-cell metrics we consider.
> > > > - **other biomedical agents** emphasize literature-based reasoning, hypothesis generation, and general biomedical Q&A, rather than end-to-end execution of single-cell pipelines with concrete code and domain-specific tool integration.
> > > > - **Recent LLM-based single-cell copilots** typically provide conversational assistance and script generation (e.g., for Scanpy), but:
> > > >   - do not separate planning, execution, and evaluation into specialized agents,
> > > >   - do not formalize a task-specific scoring function to drive pipeline optimization, and
> > > >   - do not systematically benchmark performance across dozens of datasets and multiple downstream tasks.
> > > >
> > > >
> > > > > **7. I have tried CellAgent online. Currently, it only allows the uploaded files that are less than 1GB. But lots of scRNA-seq data or single-cell-resolution ST data have larger sizes unless the analysis is based on the toy dataset. The authors should resolve this problem.、**
> > > >
> > > > Thank you for trying CellAgent online and for pointing out this practical limitation. The **1GB upload limit in our hosted demo** is purely due to server storage and bandwidth constraints and is **not** a fundamental limitation of the framework.
> > > >
> > > > CellAgent is built on top of **Scanpy/AnnData** and fully supports large-scale scRNA-seq and single-cell-resolution ST datasets, using standard strategies such as backed mode and sparse storage. In the open-source version, we provide a **local deployment option** (using Docker and configuration files) so that users can run CellAgent on their own servers without the 1GB restriction and can analyze large datasets typical of modern experiments.
> > > >
> > > > ---
> > > >
> > > > Once again, we thank you for your insightful and constructive comments. We hope that the clarifications and planned revisions described above help to better convey the specific contributions of CellAgent beyond tool orchestration, and to strengthen the rigor and transparency of our experimental evaluation.

---

### Official Review · Reviewer_FA5g · 2025-10-23

**Soundness:** 3
**Presentation:** 3
**Contribution:** 1
**Rating:** 2
**Confidence:** 5

**Summary:**

The paper presents CellAgent, an LLM-based agent for the analysis of single-cell data. The system follows a three-step analytical pipeline comprising planning, execution (including tool selection and code generation), and evaluation. The framework is implemented as a natural-language chat interface through which users can upload single-cell or spatial data and perform interactive exploration. The paper also includes a benchmark comparing the agent against other computational utilities and a baseline LLM in terms of biological task accuracy and ease of use.

**Strengths:**

Chat interface implemented at supported by the authors might be a helpful resource for people starting with single cell analysis.

**Weaknesses:**

This work is not the first attempt to automate single-cell analysis through agentic systems. Existing coding copilots such as Claude Code, OpenAI Codex, and Microsoft Copilot already support major single-cell analysis utilities, allowing users to analyze datasets locally and at lower cost. Moreover, domain-specific agents like Biomni offer broader functionality than CellAgent, while more specialized tools such as scAgent and Cell2Scale can perform key downstream analyses including cell-type annotation and exploration of cellular identity. The main distinction between Biomni and general-purpose copilots versus CellAgent lies in their flexibility—they are not constrained by a hardcoded set of tools, making them more adaptable to specific datasets and more easily extensible to new packages.

CellAgent also does not implement any guardrails to control the LLM hallucinations or any procedures to inspect the quality of generated code that could potentially compromise the quality of the findings.

**Questions:**

Below I suggest modifications that can help improve the paper.

1. Illustrate the novelty and efficiency of the proposed approach as compared to existing alternatives. How would **Cursor** on the one hand and **Biomni** perform the same task — are they worse than **CellAgent**?

2. Illustrate the robustness of **CellAgent** to LLM hallucinations. Also, how would **CellAgent** behave when prompted with a complex task that includes multiple steps?

3. The paper develops two code bases, **scOMNI** and **CellAgent**. While **scOMNI** is public, **CellAgent** is not. It is hard to evaluate the validity of the approach when the code for it is hidden. In **scOMNI**, I notice that many paths are hard-coded; this would be a major limitation to running the method locally.

4. There are several unclear aspects in the paper. For example, how do you run other methods for the benchmark? How do you compute the *AS* score for spatial domain identification in embryo, breast cancer, and kidney datasets, which do not have ground-truth data?

5. Cell-type identification is performed on a relatively easy dataset with coarse cell-type labels — extend it to more complex datasets with fine-grained labels and compare with other agentic approaches for cell typing, including a simple baseline where you give the top 50 marker genes to GPT-5 and ask it to reason over the data.

---

> ### Author Response · Authors · 2025-11-27
> **Response to Reviewer FA5g (1/4)**
>
> Dear Reviewer FA5g,
>
> We sincerely thank you for your careful and detailed review, and for recognizing the potential value of a chat-based interface for users who are new to single-cell analysis. Your comments have been very helpful in clarifying how CellAgent relates to existing copilots and biomedical agents, how we handle hallucinations and multi-step workflows, and where we need to improve the clarity of our experimental methodology and code release. We address each of your points below.
>
>
> > **1. Illustrate the novelty and efficiency of the proposed approach as compared to existing alternatives. How would Cursor on the one hand and Biomni perform the same task — are they worse than CellAgent?**
>
>
> We appreciate the suggestion. Our goal is to make clear what is **distinctive** about CellAgent relative to three main categories of existing systems:
>
> - general-purpose code copilots (Cursor, Copilot, Claude Code, Codex),
> - general-purpose biomedical scientific agents (e.g., Biomni),
> - task-specialized agents for cell typing (e.g., scAgent, Cell2Scale).
>
> **General code copilots (Cursor, Copilot, etc.).**
> Generic coding assistants can indeed call popular single-cell libraries (Scanpy, Seurat) and help users write analysis code. However, they fundamentally lack:
>
> - a **domain-specific toolkit abstraction**: they do not provide a curated, standardized space of single-cell and spatial tools with consistent interfaces and documented behavior;
> - a **closed-loop evaluation mechanism**: they do not inspect UMAPs, trajectories, or spatial patterns using domain metrics to decide whether an analysis is biologically reasonable, nor do they automatically adjust parameters when metrics or diagnostics indicate problems;
>
> In our own exploratory use of Copilot-like tools for single-cell tasks, we observed two consistent patterns: (i) performance depends strongly on the underlying model quality (strong LLM backends, such as GPT-5.1-Codex can sometimes produce good pipelines, while weaker ones (such as GPT-5-mini) often fail to run end-to-end without substantial human intervention), and (ii) evaluation is heavily dependent on the presence of ground-truth labels—without them, the copilots struggle to reliably interpret plots or identify key failure modes. In contrast, **CellAgent provides a full closed-loop workflow**: the Planner decomposes tasks, the Executor generates and executes code using sc-Omni, and the Evaluator uses domain-specific metrics and plots to autonomously select and refine pipelines. This significantly lowers the expertise required from the user and improves robustness.
>
> **General-purpose biomedical agents (Biomni).**
> We view Biomni as a highly important and complementary line of work: it aims to be a **generalist biomedical agent**, covering literature review, hypothesis generation, experimental design, and bioinformatics across many subfields (genomics, proteomics, rare disease diagnosis, etc.). Architecturally, Biomni and CellAgent share the high-level pattern of “planning + tool-based code execution.” The key differences are in **scope and depth**:
>
> - Biomni is a *broad* “biomedical generalist,” designed to operate across dozens of biomedical domains.
> - **CellAgent is a *vertical*, domain-specialized framework focused on scRNA-seq and spatial transcriptomics**, designed to behave like an experienced bioinformatics engineer rather than a generic coder.
>
> Concretely:
>
> - CellAgent covers a **full downstream pipeline** for single-cell and spatial omics—from batch correction and cell-type annotation to trajectory inference, spatial domain identification, and spatial imputation—within a single multi-agent architecture and a unified chat interface.
> - CellAgent introduces **sc-Omni**, a curated toolkit that standardizes **30+ core methods** for scRNA-seq and spatial transcriptomics into a unified, well-documented interface. This makes it possible to perform *systematic pipeline exploration and benchmarking* across more than 60 datasets, with fair comparisons across tools.
> - CellAgent incorporates a **self-reflective Evaluator** that uses task-specific metrics and diagnostic plots to score multiple candidate pipelines and automatically select and tune them **without human intervention**, providing a built-in quality-assurance layer for each task.
>
> While Biomni and similar systems are more flexible in the sense that they can, in principle, call arbitrary software, their breadth comes at the cost of the kind of deeply integrated, domain-specific evaluation loop we implement for single-cell and spatial analyses. In the revision, we will make this positioning explicit in the **Related Work** section, and we will emphasize that our goal is not to replace generalist agents like Biomni, but to provide a **default, high-fidelity pipeline** for scRNA-seq and spatial transcriptomics that labs can trust.

---

> > ### Author Response · Authors · 2025-11-27
> > **Response to Reviewer FA5g (2/4)**
> >
> > **Task-specialized agents (scAgent, Cell2Scale, etc.).**
> > Systems such as scAgent and Cell2Scale are primarily designed for **cell-type annotation and exploration of cell identity**, often across tissues and atlases. They are therefore highly complementary to CellAgent. Our design is such that these specialized methods can be wrapped as tools inside sc-Omni, allowing CellAgent to call them as powerful backends for annotation while still orchestrating the entire end-to-end workflow (from preprocessing and batch correction to spatial analysis). In the revision, we will explicitly discuss this complementarity and the possibility of integrating such agents into the sc-Omni ecosystem.
> >
> > **On “hardcoded tools” vs. flexibility.**
> > We also appreciate the concern about flexibility. While CellAgent currently builds on a **curated** set of tools in sc-Omni for robustness and reproducibility, the toolkit itself is **not hard-coded into the agent logic**. Tools are registered through a configuration-driven interface; adding a new method amounts to providing a small wrapper that exposes its inputs/outputs in the standard format. We will clarify this in the paper and in the public code, and provide examples of how researchers can add new packages to sc-Omni. Our design therefore trades “unbounded flexibility” for “highly structured, extensible, and benchmarkable” tool integration in this vertical domain.
> >
> >
> > > **2. Illustrate the robustness of CellAgent to LLM hallucinations. Also, how would CellAgent behave when prompted with a complex task that includes multiple steps?**
> >
> >
> > We fully agree that hallucinated code and unsafe execution are critical concerns for any agentic system. CellAgent incorporates several **guardrails** that specifically target these issues:
> >
> > - **Isolated execution sandbox.**
> >   All generated code is executed in a restricted Jupyter-notebook-based sandbox with no external network access and controlled file I/O. This prevents harmful side effects on the host environment and keeps all operations within a reproducible environment.
> >
> > - **Automatic self-debugging.**
> >   The Executor captures error messages, performs self-correction, and reruns code up to three times when runtime errors occur. As reported in Sec. 4.3, this self-debugging combined with planning and evaluation leads to a **96% execution success rate** across eight representative tasks, compared to only **24%** for a single-agent GPT-4 baseline that directly writes entire pipelines without such guardrails.
> >
> > - **Evaluator-driven sanity checks.**
> >   For each subtask, the Evaluator inspects task-specific quantitative metrics and diagnostic plots (e.g., UMAP structure for clustering, pseudotime continuity for trajectories, spatial coherence for domain identification). Pipelines that produce clearly unreasonable results—such as collapsed clusters, fragmented or anatomically implausible spatial domains, or degenerate trajectories—receive low scores and are discarded in favor of better candidates.
> >
> > Regarding complex tasks: all eight benchmark tasks in the paper are **multi-step workflows**, not single API calls. CellAgent always starts by having the Planner decompose the user’s natural-language query into a sequence of subtasks (e.g., quality control → normalization → dimensionality reduction → clustering → annotation), and then executes them step by step. At each stage, the Evaluator checks metrics and diagnostics before the system proceeds. If intermediate steps go wrong (e.g., over-correction in batch integration or unstable spatial clustering), the self-reflective loop triggers alternative tool choices or parameter settings.
> >
> > Motivated by your comment, we will add a concise **failure-mode analysis** in the Appendix, including concrete traces of how CellAgent responds when intermediate steps fail or produce suspicious results, and how the Evaluator and self-debugging mechanisms work together to mitigate hallucinations.

---

> > > ### Author Response · Authors · 2025-11-27
> > > **Response to Reviewer FA5g (3/4)**
> > >
> > > > **3. The paper develops two code bases, scOMNI and CellAgent. While scOMNI is public, CellAgent is not. It is hard to evaluate the validity of the approach when the code for it is hidden. In scOMNI, I notice that many paths are hard-coded; this would be a major limitation to running the method locally.**
> > >
> > > We apologize for the confusion regarding code availability. As stated in the Reproducibility Statement, the core functional code of **both** CellAgent and sc-Omni is already provided in the anonymous repository linked in the submission and will be fully open-sourced upon publication.
> > >
> > > The hard-coded paths you noticed in sc-Omni are indeed artifacts of our internal deployment and packaging. In the revised repository, we will:
> > >
> > > - replace all hard-coded paths with **configurable environment variables or YAML configuration** files;
> > > - provide **Dockerfiles and step-by-step instructions** for running CellAgent locally on user-provided datasets;
> > > - add **unit tests** to verify that each tool in sc-Omni can be invoked from a clean environment.
> > >
> > > We believe these changes will substantially improve reproducibility and make it straightforward for others to deploy and extend CellAgent locally.
> > >
> > >
> > > > **4. There are several unclear aspects in the paper. For example, how do you run other methods for the benchmark? How do you compute the AS score for spatial domain identification in embryo, breast cancer, and kidney datasets, which do not have ground-truth data?**
> > >
> > >
> > > We greatly appreciate you pointing out this detail; we provide the following clarification.
> > >
> > > **How we run other methods.**
> > > For each task, we followed the original papers and official implementations of the baselines (e.g., scVI, Harmony, Slingshot, Tangram, SpaGE), using the same or recommended hyperparameters whenever possible. Preprocessing steps such as normalization, feature selection, and batch labeling were standardized across methods as described in Appendix D, with additional method-specific preprocessing (e.g., HVG selection) applied consistently. In the revision, we will add a **summary table** listing, for each baseline, the exact command-line or Python invocation and key hyperparameters used in our experiments.
> > >
> > > **How we compute AS when domain labels are missing.**
> > >
> > > We regret that the description of the Accuracy Score (AS) was ambiguous. AS is used in our experiments to evaluate spatial imputation accuracy, not to assess spatial domain labels directly. Even in embryo, breast cancer, and kidney datasets where spatial domain annotations are unavailable, the actual expression values of the genes targeted for prediction are observed (serving as the ground truth), which allows us to rigorously evaluate imputation quality.
> > >
> > > Concretely, when the user requests spatial imputation (e.g., predicting the expression of 200 “unknown” genes), CellAgent proceeds as follows:
> > >
> > > 1. It first identifies the set of **shared highly variable genes (HVGs)** between the spatial transcriptomics dataset and a matched scRNA-seq reference (e.g., top 1000 HVGs).
> > > 2. These HVGs are split into a **training set** (e.g., 800 genes) and a **validation set** (e.g., 200 genes). The training set is used to fit candidate imputation models, and the validation set is held out as “pseudo-unobserved” ground truth.
> > > 3. Multiple imputation methods (e.g., Tangram, gimVI, SpaGE) are trained on the training genes.
> > > 4. The **Evaluator** then computes AS on the validation genes by combining four metrics—PCC, RMSE, SSIM, and JS divergence—between the predicted and true expression values on the masked entries.
> > > 5. The method with the best AS on the validation genes is selected and then applied to impute the *actual* unknown genes requested by the user.
> > >
> > > Thus, AS is only used when we have expression ground truth (even if domain labels are missing), and it is computed on held-out genes/spots created via this train–validation split procedure. For spatial domain identification on datasets lacking domain labels, we rely on datasets with known annotations (for quantitative metrics) and on expert-reviewed visualizations. We will clarify this distinction in the revised manuscript.

---

> > > > ### Author Response · Authors · 2025-11-27
> > > > **Response to Reviewer FA5g (4/4)**
> > > >
> > > > > **5. Cell-type identification is performed on a relatively easy dataset with coarse cell-type labels — extend it to more complex datasets with fine-grained labels and compare with other agentic approaches for cell typing, including a simple baseline where you give the top 50 marker genes to GPT-5 and ask it to reason over the data.**
> > > >
> > > > We appreciate your concern about the difficulty of the cell-type identification tasks. Our benchmark is not limited to datasets with coarse labels. For example, the liver dataset in our benchmark includes **27 distinct cell types**, which poses a non-trivial classification challenge and requires fine-grained discrimination among closely related subpopulations. Other datasets also contain fine-grained T-cell and myeloid subtypes beyond simple “broad lineage” labels.
> > > >
> > > > As described in the Appendix, CellAgent integrates both **database-based methods** (e.g., CellMarker2.0, ACT) and **expression-based methods** (e.g., CellTypist) and uses a GPT-4-based Evaluator to aggregate these predictions with cluster-level differentially expressed genes. In our experiments, we explicitly compare against:
> > > >
> > > > - a **pure LLM baseline** in which an advanced GPT-4-family model directly predicts cell types from cluster-level marker genes / DEGs, and
> > > > - dedicated annotation tools such as ACT, CellMarker, ScType, SCSA, CellTypist, and scGPT.
> > > >
> > > > CellAgent achieves the highest average accuracy (**0.85**) across datasets, outperforming both the pure LLM baseline and all other methods.
> > > >
> > > > ---
> > > >
> > > > Once again, we thank you for your thorough and constructive feedback. While we agree that CellAgent builds upon a growing ecosystem of LLM-based tools and agents, we hope that the clarifications and additions above (especially regarding domain-specialized evaluation, end-to-end automation, robustness, and reproducibility) help to better convey its unique value as a vertical, high-fidelity agent for single-cell and spatial transcriptomics analysis.
> > > >
> > > > Please do not hesitate to reach out if you have further questions; we are dedicated to resolving any remaining concerns to fully realize the potential of CellAgent.

---

### Official Review · Reviewer_VQec · 2025-10-31

**Soundness:** 3
**Presentation:** 3
**Contribution:** 2
**Rating:** 6
**Confidence:** 3

**Summary:**

The paper introduces CellAgent, a large language model–driven multi-agent framework for fully automated single-cell and spatial transcriptomics analysis through natural language interaction. The system integrates three agents, namely Planner, Executor, and Evaluator, operating under a self-reflective optimization loop that plans tasks, generates code, and evaluates results. A comprehensive benchmarking across over 60 datasets is also presented.

**Strengths:**

1) The study presents a technically sophisticated and well-structured multi-agent framework t
2) The inclusion of the sc-Omni toolkit provides a strong and practical foundation, consolidating many established analysis methods into a unified framework.
3) The comprehensive benchmarking on multiple downstream tasks supports the robustness and practical relevance of the method.

**Weaknesses:**

1) The biological interpretation of the outputs is limited. The evaluation focuses mainly on computational metrics without assessing whether the inferred results align with known biological mechanisms.
2) The evaluation of “human expert performance” lacks sufficient detail on the experimental design, number of participants, and reproducibility of human assessments.
3) The author do not discuss the advantages / disadvantages of their framework against some other relevant systesm, most notably SpatialAgent:
https://www.biorxiv.org/content/10.1101/2025.04.03.646459v1.abstract

**Questions:**

I would ask the authors to address the weaknesses highlighted above:
1) Please include biologically grounded evaluations, such as testing whether CellAgent recapitulates known spatial tissue structures in benchmark datasets. Walking the reader through one representative dataset out of the 60 would suffice
2) Provide more details on the human expert evaluation, including participant expertise, evaluation protocol, and inter-rater variability.
3) Please ensure that you exhaustively discuss the state-of-the-art.

---

> ### Author Response · Authors · 2025-11-27
> **Response to Reviewer VQec (1/2)**
>
> Dear Reviewer VQec,
>
> We sincerely thank you for your careful reading of the manuscript and for your positive assessment of the CellAgent framework, the sc-Omni toolkit, and the breadth of our benchmarking. Your comments are extremely helpful for strengthening the biological depth of the paper, clarifying the human expert evaluation, and better positioning CellAgent among existing systems. We address each point in detail below.
>
> > **1. The biological interpretation of the outputs is limited. The evaluation focuses mainly on computational metrics without assessing whether the inferred results align with known biological mechanisms.**
>
> We fully agree that computational metrics alone are not sufficient to establish biological validity. Beyond the quantitative benchmarks already presented, CellAgent exhibits strong biological consistency on multiple tasks, and we will make these interpretations more explicit in the revision.
>
> Representative examples include:
>
> - **PBMC cell-type annotation (Fig. 3a).**
>   On the human PBMC dataset, CellAgent refines the original annotation by subdividing CD8⁺ T cells into *naïve cytotoxic T cells* and *CD8⁺ effector T cells*. This refinement is consistent with known immunology, where naïve cytotoxic T cells represent a distinct subset that has not yet encountered antigen and remains undifferentiated into effector cytotoxic T cells.
>
> - **Hematopoietic trajectory (Fig. 3b).**
>   On the “Aging HSC Kowalczyk” dataset, CellAgent reconstructs a trajectory from LT-HSC → ST-HSC → MPP that matches the canonical differentiation hierarchy. The heatmap of the top 20 differentially expressed genes along the trajectory shows gradual up-regulation of **CD48** and **MPO**, in line with their known roles in immune cell maturation and myeloid differentiation. This supports that the inferred trajectory is not only topologically plausible but also molecularly coherent.
>
> - **DLPFC spatial domains (Fig. 4a).**
>   For human DLPFC slices, CellAgent recovers cortical layers (L1–L6) and white matter with smooth boundaries and correct ordering, in agreement with manual annotations.
>
> - **Mouse cortex imputation (Fig. 4b–d).**
>   For spatial imputation on the mouse cortex dataset, the predicted expression pattern of **Igsf21** closely matches the ground truth. Among the top spatially variable genes identified after imputation (e.g., Ptgfrn, Zyx, Rprm), the predicted laminar enrichment patterns are consistent with known cortical organization.
>
> We hope this detailed example will make it clear that CellAgent is not merely optimizing numerical scores but is able to recover biologically meaningful tissue architecture.
>
>
> > **2. The evaluation of “human expert performance” lacks sufficient detail on the experimental design, number of participants, and reproducibility of human assessments.**
>
> Thank you for pointing out that the current description of the human expert evaluation is too brief. We will clarify the experimental design, participant expertise, and reproducibility as follows.
>
> - **Participants.**
>   We recruited **10 human experts**, all with **2–5 years of experience** in single-cell and/or spatial transcriptomics analysis.
>
> - **Tasks and tools.**
>   Each expert was asked to analyze a subset of datasets and tasks drawn from the same benchmark pool used for CellAgent (including scRNA-seq and spatial analyses such as batch correction, cell-type annotation, trajectory inference, and spatial domain identification). Experts were free to use their preferred tools and workflows, primarily based on Seurat, Scanpy, and commonly used domain-specific methods.
>
> - **Procedure.**
>   For each dataset–task pair, experts received the raw or preprocessed data along with the task description, and were given a fixed time budget to complete the analysis. In parallel, CellAgent was run on the **same datasets and tasks** via natural language instructions issued by an operator, without revealing the human experts’ intermediate results. This setup allowed us to compare CellAgent and human experts under comparable conditions.
>
> - **Blinded scoring and reproducibility.**
>   The final outputs (processed data, visualizations, and written conclusions) from both humans and CellAgent were anonymized and randomly shuffled. They were then evaluated by an independent panel of **5 additional experts** who did not participate in the original analyses. These evaluators were asked to assign **0–10 quality scores** under a blinded setting, based on criteria such as correctness, biological interpretability, and overall presentation. In the revision, we will describe this protocol in detail in the Appendix and report inter-rater agreement statistics (using standard reliability measures) to quantify the reproducibility of the human assessments.
>
> We believe these additions will make the human expert evaluation component much more transparent and reproducible.

---

> > ### Author Response · Authors · 2025-11-27
> > **Response to Reviewer VQec (2/2)**
> >
> > > **3. The author do not discuss the advantages / disadvantages of their framework against some other relevant systesm, most notably SpatialAgent**
> >
> >
> > We appreciate you highlighting SpatialAgent. SpatialAgent is indeed an important system: it is a fully autonomous AI agent for **spatial biology**, integrating LLMs with dynamic tool execution and adaptive reasoning to support key stages such as experimental design, multimodal spatial data analysis, and hypothesis generation in spatial transcriptomics.
> >
> > In the revision, we will add detailed discussion in the **Related Work** section to discuss SpatialAgent in depth and to position CellAgent relative to it and other recent systems. In particular:
> >
> > - **Scope and focus.**
> >   SpatialAgent is designed specifically for spatial-biology workflows, spanning experiment design, multi-scale spatial annotation, and downstream spatial analyses. In contrast, **CellAgent is a domain-specialized framework that unifies both scRNA-seq and spatial transcriptomics** within a single multi-agent architecture, and focuses on fully automating a broad downstream workflow, including batch correction, cell-type annotation, trajectory inference, spatial domain identification, and spatial imputation via natural language interaction.
> >
> > - **Tooling and standardization.**
> >   CellAgent introduces **sc-Omni**, a curated toolkit that standardizes **30+ core methods** in single-cell and spatial omics into a unified, well-documented interface. This design enables fair comparison among tools, systematic exploration of alternative pipelines, and extensive benchmarking across more than 60 datasets.
> >
> > - **Self-reflective evaluation and autonomous optimization.**
> >   Beyond executing individual analyses, CellAgent incorporates a **self-reflective Evaluator** that uses domain-specific metrics and visual diagnostics to perform multi-candidate scoring and selection. This feedback loop enables **fully automated hyperparameter and pipeline optimization without human intervention**, providing an additional quality-assurance layer on top of the sc-Omni toolkit.
> >
> > At the same time, we will acknowledge that SpatialAgent has strengths in experimental design and multimodal integration that are complementary to our focus on unified downstream analysis and benchmarking. Beyond SpatialAgent, we will also expand our discussion of other state-of-the-art systems such as GPTCellType, AutoBA, Biomni, and related agentic frameworks for omics data (see response to Reviewer ia61). We will explicitly summarize the advantages and limitations of CellAgent relative to these systems.
> >
> > ---
> >
> > Once again, we thank you for your constructive feedback and for your positive assessment of the technical sophistication and practical relevance of CellAgent. Your suggestions have helped us substantially strengthen the biological interpretation, clarify our human expert evaluation, and situate CellAgent more clearly within the rapidly evolving landscape of LLM-based agents for single-cell and spatial biology.

---

### Official Review · Reviewer_szPk · 2025-11-09

**Soundness:** 2
**Presentation:** 3
**Contribution:** 2
**Rating:** 2
**Confidence:** 5

**Summary:**

This paper introduces CellAgent, a multi-agent framework that uses LLMs organized into three biological Planner, Executor, and Evaluator to automate end-to-end scRNA-seq analysis, including preprocessing, batch correction, cell-type annotation, and trajectory inference. CellAgent combines a curated toolkit sc-Omni of domain methods, runs generated code in a notebook sandbox, and applies a self-reflective optimization loop.

**Strengths:**

1. CellAgent automates single-cell and spatial analyses via natural language, which is a meaningful system that has clear practical value.

2. The paper includes rich biological visualizations, enhances interpretability, and aligns with standard practices in single-cell and spatial omics.

**Weaknesses:**

1. CellAgent proposes an empirical engineering system by combining existing tools, but fails to elevate the framework into a principal scientific formulation. It does not clearly articulate why or under what principles the multi-agent coordination works.

2. There is only one type of  LLM (GPT-4o) that both interprets results and decides which algorithm is best, which raises the risk of self-circularity. Masking algorithm IDs alone do not guarantee fairness. The absence of human-expert-LLM-related validation or multi-LLMs consensus undermines the robustness and objectivity of the reported performance claims.

3. It’s unclear whether the framework ensures scientific validity, or simply produces metric scores and visual plots. Since it lacks expert-reviewed case studies or even a single example where the agent detects and corrects a biologically unreasonable output. In single-cell analysis, the former is not enough.

4. The paper lacks ablation studies to justify the necessity of each component, or based on different LLM APIs apart from GPT-4o. Only plainly stating that it simulates the workflow of deep-thinking is not reasonable.

**Questions:**

Could you release exact dataset lists (accession IDs), all preprocessing scripts or methods with random seeds?

Could you specify the configuration or computational cost of the LLM components, including key parameters (e.g., temperature, tokens, API version, number of calls per task) and the estimation of the total inference cost and runtime for the experiments?

What is the success rate of CellAgent? What will happen if it chooses the wrong tool? Have you analyzed the types of tasks or datasets where CellAgent fails or produces suboptimal pipelines?

---

> ### Author Response · Authors · 2025-11-27
> **Response to Reviewer szPk (1/5)**
>
> Dear Reviewer szPk,
>
> We sincerely thank you for your thoughtful and detailed review, and for recognizing both the practical value and the biological professionalism of CellAgent. Our goal is to lower the barrier to high-quality single-cell and spatial transcriptomics analysis while adhering to rigorous scientific standards. We particularly appreciate your constructive criticism regarding the scientific formulation of the multi-agent framework and the rigor of our evaluation process. Your insightful comments have prompted us to conduct extensive new experiments—including multi-LLM validation and component-wise ablations—which we believe clarify the design motivation and improve the empirical support of our framework.
>
> Below, we address your concerns point-by-point and outline the new experiments and clarifications.
>
> > **1. CellAgent proposes an empirical engineering system by combining existing tools, but fails to elevate the framework into a principal scientific formulation. It does not clearly articulate why or under what principles the multi-agent coordination works.**
>
> We fully agree that CellAgent should not be viewed merely as a collection of tools. In the revision, we will make explicit that we formulate single-cell and spatial analysis as a *hierarchical decision process* over the space of analysis pipelines.
>
> Concretely, CellAgent decomposes an analysis session into three explicit sub-problems:
>
> 1. **Planning / task decomposition** – mapping a natural-language request and dataset summary into an ordered list of subtasks;
> 2. **Candidate pipeline generation** – for each subtask, selecting tools from sc-Omni and generating executable code;
> 3. **Biology-aware selection and refinement** – evaluating multiple candidate runs using task-specific metrics and visual diagnostics, and selecting the best pipeline.
>
> These three stages are handled respectively by the **Planner**, **Executor**, and **Evaluator**, and together induce a principled *black-box optimization* view over a discrete space of pipelines, rather than ad-hoc orchestration. This architecture is motivated by a growing consensus in the agent literature that multi-agent specialization improves reliability on complex, role-structured tasks (e.g., MetaGPT, AutoGen, CAMEL), where each agent can focus on a narrower profile and reduce hallucinations.
>
> To guide high-quality collaboration, CellAgent follows two explicit principles:
>
> - **Hierarchical decomposition.** We separate “understanding & planning”, “tool invocation & code execution”, and “result evaluation”. This lowers prompt complexity for each agent and allows us to inject domain-specific constraints and metrics into each stage in a modular way.
> - **Self-reflective optimization.** At key subtasks, the Evaluator drives a “multi-candidate → scoring → selection” loop, enabling local search and correction instead of a single forward pass.
>
> Empirically, this architecture substantially improves robustness. As shown in Sec. 4.3 and Fig. 5, a naïve single-agent baseline where GPT-4 directly writes the entire pipeline (“GPT-4”) achieves only **24%** execution success across eight representative tasks, whereas **CellAgent@3** (with Planner–Executor–Evaluator and self-optimization) reaches **96%** success and slightly higher expert-rated quality. In the revision we will make this “single-agent vs. multi-agent” comparison explicitly, and we will clarify the decision-process perspective in the Methods section.
>
> > **2. There is only one type of LLM (GPT-4o) that both interprets results and decides which algorithm is best, which raises the risk of self-circularity. Masking algorithm IDs alone do not guarantee fairness. The absence of human-expert-LLM-related validation or multi-LLMs consensus undermines the robustness and objectivity of the reported performance claims.**
>
> We appreciate this concern regarding self-circularity. We would like to clarify that our evaluation mechanism is designed to minimize bias, and we have added new experiments to demonstrate its robustness.

---

> > ### Author Response · Authors · 2025-11-27
> > **Response to Reviewer szPk (2/5)**
> >
> > **(a) Design choices that reduce bias.**
> > Even in the original submission, the Evaluator was *not* simply asked to “pick whatever it likes”:
> > - All algorithm identifiers are **masked** during evaluation, so the model cannot exploit name recognition or training-data popularity of specific tools.
> > - Evaluation criteria are **task-specific and multi-modal**, combining:
> >   - quantitative metrics (e.g., batch correction scores, AS for imputation, ARI for clustering / spatial domains),
> >   - when available, metrics directly tied to ground-truth labels,
> >   - and visual assessments guided by domain knowledge (for trajectory continuity, spatial domain coherence, etc.).
> > - For some tasks—most notably **spatial transcriptomics imputation**—the Evaluator relies almost entirely on *quantitative* metrics (AS and its components), without visual assessment.
> > - The memory of the Executor and Evaluator is **strictly separated**: the Evaluator only sees final outputs and metrics for each candidate run, but not the code generation process, which reduces the risk of “justifying its own prior choices”.
> > - Finally, Sec. 4.3 includes a **human–LLM validation**: independent human experts scored the quality of the final pipelines based on standard visualizations and metrics, and CellAgent achieved slightly higher average scores than human experts while being faster.
> >
> > **(b) Multi-LLM robustness.**
> > Motivated directly by your comment, we instantiated CellAgent with different backbone models for all agents while keeping sc-Omni and prompts fixed. Specifically, we compared GPT-4o, Qwen3-VL-235B, and Qwen3-Omni-30B across five downstream tasks. The average scores remain highly consistent:
> >
> > | Backbone            | Batch correction | Cell-type annotation | Trajectory inference | Spatial domains | Spatial imputation |
> > |---------------------|:----------------:|:--------------------:|:--------------------:|:---------------:|:------------------:|
> > | GPT-4o              |       0.67       |         0.85         |         0.50         |      0.47       |        0.88        |
> > | Qwen3-VL-235B       |       0.65       |         0.85         |         0.44         |      0.46       |        0.88        |
> > | Qwen3-Omni-30B      |       0.60       |         0.80         |         0.40         |      0.40       |        0.87        |
> >
> > The similar ranking of tools and the close scores across three strong models suggest that the selection is primarily driven by *task-specific metrics* rather than idiosyncratic preferences of a single backbone.
> >
> > **(c) Ablation of the Evaluator.**
> > To further disentangle execution robustness from LLM-based evaluation, we ran a variant **without the Evaluator**, in which CellAgent executes a single candidate pipeline per step without self-reflective optimization. The average scores across the five tasks are:
> >
> > | Method                   | Batch correction | Cell-type annotation | Trajectory inference | Spatial domains | Spatial imputation |
> > |---------------------|:----------------:|:--------------------:|:--------------------:|:---------------:|:------------------:|
> > | CellAgent (GPT-4o)       |       0.67       |         0.85         |         0.50         |      0.47       |        0.88        |
> > | CellAgent w/o Evaluator  |       0.60       |         0.75         |         0.20         |      0.35       |        0.75        |
> >
> > This shows that:
> >
> > 1. Even without Evaluator, the execution pipeline—driven by Planner + Executor + sc-Omni—is already quite robust and competitive with task-specific baselines.
> > 2. The Evaluator is crucial for difficult tasks such as trajectory inference and spatial analysis, where the overall score drops from 0.50 to 0.20 when self-reflection is disabled. This decline occurs because, without the Evaluator, the system defaults to the first runnable solution regardless of quality. Consequently, poor or biologically implausible results are accepted without inspection, which substantially undermines the scientific meaningfulness of the analysis.

---

> > > ### Author Response · Authors · 2025-11-27
> > > **Response to Reviewer szPk (3/5)**
> > >
> > > > **3. It’s unclear whether the framework ensures scientific validity, or simply produces metric scores and visual plots. Since it lacks expert-reviewed case studies or even a single example where the agent detects and corrects a biologically unreasonable output. In single-cell analysis, the former is not enough.**
> > >
> > > Your emphasis on going beyond merely generating plots to ensuring scientific validity is extremely important to us. We fully agree that metric-only evaluation is insufficient in single-cell analysis; biological plausibility and expert interpretation are essential. Our experiments were explicitly designed to test biological validity, and several case studies already appear in the current submission. We will highlight them more clearly and expand the narrative in the revision.
> > >
> > > Representative examples include:
> > >
> > > - **PBMC cell-type annotation (Fig. 3a).**
> > >   On the human PBMC dataset, CellAgent refines the original annotation by subdividing CD8⁺ T cells into *naïve cytotoxic T cells* and *CD8⁺ effector T cells*. This refinement is consistent with known immunology, where naïve cytotoxic T cells represent a distinct subset that has not yet encountered antigen and remains undifferentiated into effector cytotoxic T cells.
> > >
> > > - **Hematopoietic trajectory (Fig. 3b and Appendix Fig. 12).**
> > >   On the “Aging HSC Kowalczyk” dataset, CellAgent reconstructs a trajectory from LT-HSC → ST-HSC → MPP that matches the canonical differentiation hierarchy. The heatmap of the top 20 differentially expressed genes along the trajectory shows gradual up-regulation of **CD48** and **MPO**, in line with their known roles in immune cell maturation and myeloid differentiation. This supports that the inferred trajectory is not only topologically plausible but also molecularly coherent.
> > >
> > > - **DLPFC spatial domains (Fig. 4a).**
> > >   For human DLPFC slices, CellAgent recovers cortical layers (L1–L6) and white matter with smooth boundaries and correct ordering, in agreement with manual annotations.
> > >
> > > - **Mouse cortex imputation (Fig. 4b–d).**
> > >   For spatial imputation on the mouse cortex dataset, the predicted expression pattern of **Igsf21** closely matches the ground truth. Among the top spatially variable genes identified after imputation (e.g., Ptgfrn, Zyx, Rprm), the predicted laminar enrichment patterns are consistent with known cortical organization.
> > >
> > > We hope this will better demonstrate that CellAgent goes beyond producing “nice plots” and actively enforces biological validity.
> > >
> > > > **4. The paper lacks ablation studies to justify the necessity of each component, or based on different LLM APIs apart from GPT-4o. Only plainly stating that it simulates the workflow of deep-thinking is not reasonable.**
> > >
> > > We appreciate the request for more systematic ablations. Building on your comment, we have now performed and report the following additional ablations:
> > >
> > > - **Alternative backbone LLMs.**
> > >   We instantiated CellAgent with GPT-4o, Qwen3-VL-235B, and Qwen3-Omni-30B. The resulting scores are comparable, indicating that CellAgent’s performance is not tied to a specific API and can transfer well across different strong LLM backbones.
> > >
> > >   | Backbone            | Batch correction | Cell-type annotation | Trajectory inference | Spatial domains | Spatial imputation |
> > >   |---------------------|:----------------:|:--------------------:|:--------------------:|:---------------:|:------------------:|
> > >   | GPT-4o              |       0.67       |         0.85         |         0.50         |      0.47       |        0.88        |
> > >   | Qwen3-VL-235B       |       0.65       |         0.85         |         0.44         |      0.46       |        0.88        |
> > >   | Qwen3-Omni-30B      |       0.60       |         0.80         |         0.40         |      0.40       |        0.87        |
> > >
> > >
> > > - **Without sc-Omni and multi-agent coordination.**
> > >   In this variant, the Executor relies solely on its internal knowledge to write code, without access to the curated sc-Omni toolkit or the other agents. This effectively reduces the system to plain GPT-4 code generation. Our preliminary experiments show a much higher rate of runtime errors and frequent calls to outdated or non-existent libraries, leading to substantial degradation in overall performance compared with the full CellAgent framework.
> > >
> > > - **Without hierarchical planning (Planner ablation).**
> > >   Removing the Planner reduces CellAgent to a single-step code generator with access to sc-Omni tools. Across eight representative tasks, the execution success rate drops notably, especially on multi-step spatial workflows, where misordering of preprocessing and downstream analyses is common.
> > >
> > >   | Method |1|2|3|4|5|6|7|8|
> > >   |-----------------------------|:------:|:------:|:------:|:------:|:------:|:------:|:------:|:------:|
> > >   | CellAgent w/ Planner  |  0.90  |  1.00  |  0.95  |  0.95  |  1.00  |  1.00 | 0.95 | 0.95 |
> > >   | CellAgent w/o Planner |  0.75  |  0.92  |  0.76  |  0.80  |  0.70  |  0.90 | 0.80 | 0.80 |

---

> > > > ### Author Response · Authors · 2025-11-27
> > > > **Response to Reviewer szPk (4/5)**
> > > >
> > > > - **Without self-reflection (Evaluator ablation).**
> > > >   Disabling the iterative optimization and selection mechanism leads to a substantial drop in overall scores, particularly for trajectory inference (from 0.50 to 0.20) and spatial tasks. This demonstrates the effectiveness of the Evaluator as a quality-enhancement and safety layer on top of an already robust execution pipeline.
> > > >
> > > > | Method                   | Batch correction | Cell-type annotation | Trajectory inference | Spatial domains | Spatial imputation |
> > > > |--------------------------|:----------------:|:--------------------:|:--------------------:|:---------------:|:------------------:|
> > > > | CellAgent (GPT-4o)       |       0.67       |         0.85         |         0.50         |      0.47       |        0.88        |
> > > > | CellAgent w/o Evaluator  |       0.60       |         0.75         |         0.20         |      0.35       |        0.75        |
> > > >
> > > >
> > > > - **Without memory optimization (memory ablation).**
> > > >   We ablated the dual global–local memory architecture. With memory optimization, GPT-4o and Qwen3-VL-235B maintain high execution success rates across all eight tasks (mostly in the 0.90–1.00 range), and even the weaker Qwen3-Omni-30B backbone remains between 0.60 and 0.75. Removing memory consistently degrades performance for all three backbones, with the largest drops observed for Qwen3-Omni-30B (down to 0.40–0.65 on several tasks). These results confirm that selectively storing only finalized code in global memory and using short-lived local memory improves both stability and token efficiency.
> > > >
> > > >
> > > > **With memory optimization**
> > > >
> > > > | Model                       | Task 1 | Task 2 | Task 3 | Task 4 | Task 5 | Task 6 | Task 7 | Task 8 |
> > > > |-----------------------------|:------:|:------:|:------:|:------:|:------:|:------:|:------:|:------:|
> > > > | CellAgent (GPT-4o)          |  0.90  |  1.00  |  0.95  |  0.95  |  1.00  |  1.00  |  0.95  |  0.95  |
> > > > | CellAgent (Qwen3-Omni-30B)  |  0.75  |  0.65  |  0.65  |  0.65  |  0.60  |  0.60  |  0.70  |  0.60  |
> > > > | CellAgent (Qwen3-VL-235B)   |  0.90  |  1.00  |  0.95  |  0.95  |  1.00  |  1.00  |  1.00  |  0.95  |
> > > >
> > > > **Without memory optimization**
> > > >
> > > > | Model                       | Task 1 | Task 2 | Task 3 | Task 4 | Task 5 | Task 6 | Task 7 | Task 8 |
> > > > |-----------------------------|:------:|:------:|:------:|:------:|:------:|:------:|:------:|:------:|
> > > > | CellAgent (GPT-4o)          |  0.80  |  0.85  |  0.85  |  0.80  |  0.90  |  0.90  |  0.85  |  0.85  |
> > > > | CellAgent (Qwen3-Omni-30B)  |  0.65  |  0.45  |  0.45  |  0.40  |  0.40  |  0.40  |  0.50  |  0.40  |
> > > > | CellAgent (Qwen3-VL-235B)   |  0.77  |  0.80  |  0.85  |  0.75  |  0.80  |  0.80  |  0.85  |  0.85  |
> > > >
> > > >
> > > >
> > > >
> > > > **5. Could you release exact dataset lists (accession IDs), all preprocessing scripts or methods with random seeds?**
> > > >
> > > > Thank you for emphasizing reproducibility. In Appendix F we provided an overview of all datasets and their original references. In the Reproducibility Statement we also state that the complete sc-Omni toolkit and core functional code will be made publicly available via an anonymous repository.
> > > >
> > > > In addition, we do not apply task-specific manual preprocessing beyond basic quality control; instead, we delegate these steps entirely to CellAgent. This design better reflects the intended user experience: users can provide raw or minimally processed data, and CellAgent will automatically infer the appropriate preprocessing operations based on the dataset, explicitly list them in its planned steps, and execute them in order.
> > > >
> > > > The anonymous code repository we submitted already includes end-to-end examples of such workflows. We will make this clearer in the revision to further strengthen the reproducibility statement.

---

> > > > > ### Author Response · Authors · 2025-11-27
> > > > > **Response to Reviewer szPk (5/5)**
> > > > >
> > > > > > **6. Could you specify the configuration or computational cost of the LLM components, including key parameters (e.g., temperature, tokens, API version, number of calls per task) and the estimation of the total inference cost and runtime for the experiments?**
> > > > >
> > > > > We appreciate this suggestion and agree that these details should be clearly documented. In our experiments:
> > > > >
> > > > > - We fix **temperature = 0.0** and **top_p = 1.0** for all calls to ensure deterministic behavior and reproducibility.
> > > > > - For each analysis session:
> > > > >   - the **Planner** is called at most 3 times (initial plan + at most two revisions);
> > > > >   - the **Executor** is invoked once per subtask, with up to 3 self-debug iterations per step when runtime errors occur;
> > > > >   - the **Evaluator** is called whenever a candidate run completes successfully, and when multiple candidates are compared.
> > > > > - On our benchmark suite, a complete CellAgent pipeline takes on average about **8 minutes** of wall-clock time, compared to **13 minutes** for human experts (Sec. 4.3, Fig. 5).
> > > > > - Under the current GPT-4o pricing, the **LLM usage cost per dataset** is approximately **1–3 USD**, depending on task complexity and number of self-debug iterations.
> > > > >
> > > > > In the revision, we will add a dedicated subsection in the experimental setup (“LLM configuration and computational cost”) summarizing these parameters, alongside CPU/GPU usage where relevant. We also note that the agent framework itself is not tied to any particular hardware.
> > > > >
> > > > > > **7. On success rate, behavior under wrong tool choices, and failure modes**
> > > > >
> > > > > As reported in Sec. 4.3 and the introduction to the human-centered evaluation, **CellAgent@3** achieves an average **execution success rate of 96%** across eight representative tasks and more than 60 datasets. We define a “successful execution” as a pipeline that:
> > > > >
> > > > > 1. completes without runtime errors in the sandbox; and
> > > > > 2. produces *semantically valid* outputs for all subtasks (e.g., non-empty clusters, a valid trajectory graph, well-formed spatial domains).
> > > > >
> > > > > Regarding wrong tool choices, at each step CellAgent’s self-reflective optimization mechanism runs **multiple candidate tools or configurations**. For each candidate, the Evaluator aggregates task-specific quantitative metrics (e.g., batch correction score, ARI, AS) and visual coherence into a score. If a tool produces clearly poor outputs—e.g., strong over-correction of batches, spatial domains that violate tissue boundaries, or trajectories with fragmented pseudotime—the Evaluator assigns it a low score and it will not be selected. When all candidates perform similarly (e.g., particularly challenging datasets), CellAgent still selects the relatively best option, but we surface the final scores in the logs so that users can judge the reliability of the result.
> > > > >
> > > > > We agree that a more systematic analysis of failure modes is valuable. CellAgent is not perfect, and motivated by your comment we have added a new “failure mode analysis” section in the Appendix. The main failure modes we observe are:
> > > > >
> > > > > 1. **Extremely low-quality data**, such as severe dropout or very small sample sizes, where no tool can extract stable biological structure, leading to noisy clusters or trajectories.
> > > > > 2. **Highly ambiguous or contradictory user instructions**, which may cause the Planner to misinterpret the task and, within the limited retry budget, fail to recover the intended workflow.
> > > > > 3. **Conflicting objectives** (e.g., aggressive batch effect removal vs. preserving subtle biological differences), where the Evaluator may settle on a locally optimal but sub-ideal trade-off.
> > > > >
> > > > > We will describe these scenarios so that users better understand the current limits and behavior of CellAgent on difficult datasets.
> > > > >
> > > > >
> > > > > ----
> > > > >
> > > > > Once again, we thank you for your insightful comments and for acknowledging the practical relevance of CellAgent. We believe that the additional clarifications and new experiments described above will significantly strengthen the conceptual and empirical foundation of the framework, and we are grateful for the opportunity to improve the manuscript accordingly. We would also greatly appreciate any further suggestions on analyses or clarifications that you feel would help strengthen the paper.

---

### Author Response · Authors · 2025-11-28
**Summary of Updates**

Dear Reviewers,

We would like to express our deepest and most sincere thanks to all of you for your thoughtful, detailed, and highly constructive reviews, as well as for the encouraging comments on the completeness of the CellAgent framework, the breadth of experiments, and its practical value for single-cell and spatial transcriptomics analyses. Your feedback has been invaluable for improving this work. We look forward to engaging in further discussion, and we are fully committed to addressing any concerns during the discussion period.

During the discussion period, the main changes we have made include:

1. **New ablations on backbone models, memory, and the Evaluator.**
   We added a series of experiments to assess CellAgent’s robustness across different LLM backbones and architectural components. Specifically, we now compare:

   * CellAgent instantiated with **GPT-4o**, **Qwen3-VL-235B**, and **Qwen3-Omni-30B**,
   * with and without the **dual memory optimization** (measuring multi-step execution success rates), and
   * with and without the **Evaluator** (measuring downstream task metrics on batch correction, cell-type annotation, trajectory inference, spatial domain identification, and spatial imputation).

     These results show that (i) CellAgent’s behavior is consistent across strong backbones, (ii) memory optimization significantly improves execution stability, and (iii) the Evaluator provides substantial gains on the more challenging downstream tasks.

2. **New comparisons against integrated / agentic frameworks.**
   In addition to the individual tools (scVI, Harmony, Slingshot, Tangram, SpaGE, scGPT, etc.) reported in the original version, we have now systematically compared CellAgent to several integrated or agentic frameworks, on five key downstream tasks (batch correction, cell-type annotation, trajectory inference, spatial domain identification, and spatial imputation).

3. **Clarifications and additional analyses across multiple aspects of the paper.**
   In response to comments from several reviewers, we have:

   * **Clarified the conceptual novelty**.
   * **Detailed the human expert evaluation protocol**.
   * **Expanded the biological interpretation and case studies**.
   * **Improved fairness and transparency in benchmarking**.
   * **Substantially expanded the Related Work section**.

In addition, we have **updated the anonymous code repository** to improve reproducibility and accessibility.
   The anonymous GitHub repository linked in the submission now contains:

   * cleaned-up versions of the **sc-Omni** toolkit and the **core CellAgent logic**, and
   * a new **from-scratch tutorial for batch correction**, which walks through how the Evaluator:

     * computes quantitative integration metrics via explicit rules,
     * combines them with visual assessments (via GPT-4o) into a weighted score, and
     * uses this score to select the most appropriate batch-correction method across different datasets.
       This example is intended to concretely demonstrate how CellAgent evaluates and gates pipelines to maintain robustness across heterogeneous data.


We kindly invite all reviewers to take a look at these revisions and new results. If there are any remaining concerns, questions, or suggestions, we are happy to discuss them in depth and will do our best to further improve this work during the discussion period. Thank you again for your time and careful evaluations.

The authors

---

### Author Response · Authors · 2025-12-03
**Rebuttal Summary**

Dear Area Chair,

We sincerely thank all reviewers for their detailed evaluations and constructive feedback. We are also very grateful to you for, under the current discussion mechanism, synthesizing a large amount of material and making an integrated judgment with limited direct author–reviewer interaction. We briefly summarize below the main changes in the revised version and how they address the key concerns.

We are very appreciative of the positive feedback:

- **szPk**: CellAgent “automates single-cell and spatial analyses via natural language, which is a meaningful system that has clear practical value,” with “rich biological visualizations” and alignment with standard single-cell/spatial practices.
- **VQec**: Highlights a “technically sophisticated and well-structured multi-agent framework,” a solid sc-Omni toolkit, and comprehensive benchmarking across many tasks and datasets.
- **FA5g**: Notes that the chat interface “might be a helpful resource for people starting with single cell analysis,” with potential community value.
- **ia61**: Emphasizes a “timely idea” and “practical integration of many standard tools,” with comprehensive experiments and performance better than compared methods (“soundness/presentation = 4, rating = 6”).

We are very grateful for these encouraging comments. During the discussion period, we carefully studied and responded to all key questions raised by the reviewers, conducted substantial additional experiments and analyses, and thoroughly examined and compared the related systems highlighted by the reviewers. While practical constraints prevented extended one-on-one discussions with each reviewer, we sincerely hope that the added experiments, clarifications, and rewriting have substantially improved the paper and alleviated concerns regarding novelty, rigor, and reproducibility.

Below we summarize our responses to several key points in a point-wise manner.

---

### 1. Conceptual framing and self-reflective optimization
- **Raised by:** szPk, ia61, FA5g
- **What we changed:**
  - Explicitly formulate CellAgent as a hierarchical decision / black-box optimization process over discrete analysis pipelines.
  - Clarify that the multi-agent pattern is a general paradigm, and emphasize our domain-specific contributions.
  - Clarify that self-reflective optimization is an inference-time multi-candidate search + task-specific scoring + selection mechanism.
- **Where:** Section 2 (Related Work), Section 3 (Methods), Appendix E.1–E.2 (overall comparison and comparison with other LLM-driven agents), Appendix C.6.

### 2. Ablations, baselines, and fairness of comparisons
- **Raised by:** szPk, ia61, FA5g
- **What we changed:**
  - Add additional ablations.
  - Add quantitative comparisons with other single-cell / agentic frameworks.
  - Make explicit that baselines follow their original or community best practices.
- **Where:** New Figure 7 (comparison with other single-cell analysis agents), new Figure 8 (memory mechanism across base models), expanded experimental section.

### 3. Self-circularity and fairness of evaluation
- **Raised by:** szPk, ia61
- **What we changed:**
  - Describe in more detail how we mitigate evaluation bias (ID masking, metric-based evaluation, separation of Executor/Evaluator memory, multi-backbone consistency checks).
  - Clarify the blind human evaluation protocol used to reduce bias.
- **Where:** Section 3.2, Appendix C.6 (Evaluator), Figure 15 and surrounding text (error bars and protocol), Appendix B.2 (human evaluation details), Appendix C.5 (AS metric), Appendices E.2–E.3 (multi-backbone experiments).

### 4. Biological validity
- **Raised by:** szPk, ia61, VQec
- **What we changed:**
  - Expand the biological narrative for representative case studies, emphasizing consistency with known biology.
- **Where:** Results section around these case studies (extended descriptions).

### 5. 1GB limit in the online demo
- **Raised by:** ia61
- **What we changed:**
  - Thank the reviewer for taking the time to test the online demo. Clarify that the 1GB upload limit is an engineering constraint of the hosted demo, not of the framework itself, and point users to local deployment.

---

In summary, we are very grateful for the time and effort that the reviewers and you have invested in evaluating and discussing our submission. We have taken each critique seriously and made substantive changes through additional experiments, more formal descriptions, richer biological case studies, stronger baselines and ablations, and improved code and documentation. Our aim is to make CellAgent conceptually clearer, and practically more useful for the single-cell and spatial transcriptomics community.

We sincerely hope that these revisions adequately address the reviewers’ concerns and assist you in making your final decision.

With sincere thanks,

The authors

---

### Meta-Review · Area_Chair_ZXkM · 2026-01-04

**Summary:**

CellAgent introduces large language models (LLMs) and a multi-agent collaboration mechanism into single-cell and spatial transcriptomics data analysis, allowing users to bypass complex analytical steps and run an entire bioinformatics pipeline simply by entering instructions or plain text.

**Reviewer Concerns:**

Multi-LLM validation, ablation studies, cost breakdowns, and failure-mode analyses have been settled, yet “scientific formalization” and “independent expert verification” remain unaddressed, while CellAgent’s computational cost and complexity are still far from negligible.

**Reviewer Scores:**

The authors provided a detailed rebuttal; although a few issues remain unresolved, I believe most reviewers are likely inclined to raise their scores.

---

### Decision · Program_Chairs · 2026-01-26

Accept (Poster)